# VCC: Scaling Transformers to 128K Tokens or More by Prioritizing Important Tokens

**Zhanpeng Zeng**
University of Wisconsin, Madison
zzeng38@wisc.edu

**Cole Hawkins**
AWS AI
colehawk@amazon.com

**Mingyi Hong**
University of Minnesota, Minneapolis & AWS AI
mhong@umn.edu

**Aston Zhang**
AWS AI
astonz@amazon.com

**Nikolaos Pappas**
AWS AI
nppappa@amazon.com

**Vikas Singh**
University of Wisconsin, Madison
vsingh@biostat.wisc.edu

**Shuai Zheng**
AWS AI
shzheng@amazon.com

## Abstract

Transformers are central in modern natural language processing and computer vision applications. Despite recent works devoted to reducing the quadratic cost of such models with respect to sequence length, dealing with ultra long sequences (e.g., >16K tokens) remains challenging. Applications such as answering questions based on a book or summarizing a scientific article are inefficient or infeasible. Here, we propose to significantly improve the efficiency of Transformers for ultra long sequences, by compressing the sequence into a much smaller representation at each layer. Specifically, by exploiting the fact that in many tasks, only a small subset of special tokens, which we call VIP-tokens, are most relevant to the final prediction, we propose a VIP-token centric compression (VCC) scheme which selectively compresses the sequence based on their impact on approximating the representation of the VIP-tokens. Compared with competitive baselines, our algorithm is not only efficient (achieving more than $3\times$ compute efficiency gain compared to baselines on 4K and 16K lengths), but also offers competitive/better performance on a large number of tasks. Further, we show that our algorithm scales to 128K tokens (or more) while consistently offering accuracy improvement. Code is available at `https://github.com/mlpen/VCC`.

## 1 Introduction

The Transformer [32] is a fundamental/foundational architecture for natural language processing (NLP) and computer vision. It has shown remarkable performance across NLP applications including machine translation [32], language inference [10], and summarization [14]. Transformers have also been successfully applied to various visual recognition tasks and achieve impressive results [11, 4, 41]. Unfortunately, the runtime/memory needs of Transformers involve an unfavorable dependence on the input sequence length, making the use of Transformers for ultra-long sequence applications difficult. Therefore, many studies on Transformers make use of strategies such as truncation to ensure that the input sentence length is at most 512, e.g., BERT, T5, and other Transformer-based language models [36, 22, 27]. Unfortunately, such a truncation, and other related strategies, inevitably results in loss of accuracy, the extent of which can vary from one task/dataset to another. Consequently, improving the efficiency for longer input sequence length is a key focus of many proposals. These developments are

37th Conference on Neural Information Processing Systems (NeurIPS 2023).

important milestones, and they have reduced the quadratic dependency on sequence lengths to linear [6, 25, 35, 2, 38, 40, 39]. Currently, many Transformer models can process samples with sequence lengths of up to 4K (and even 16K). Very recently, results of newer models being able to handle much longer sequences have appeared [3].

**Rationale.** It is natural to ask whether the ability to process longer sequences is worth the trouble. The short answer is yes. Improved accuracy has been reported on long sequence tasks [2, 38, 14]. So, what is stopping us from harvesting even stronger gains in accuracy by feeding even longer sequences to such models? Models such as Longformer [2] and Big Bird [38] become slow and consume an excessive amount of memory as the sequence length keeps increasing. See Fig. 1 for illustration. Why? The representation update of every token involves computing

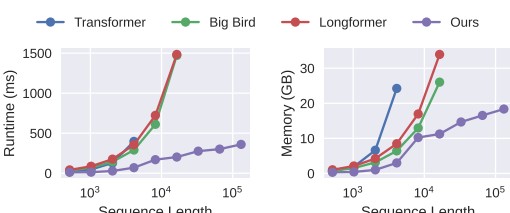

Figure 1: Model efficiency of processing one sequence on a NVIDIA A100 as sequence length increases (note logarithm $x$ axis).

efficient attention and feed-forward network at each layer. This incurs a linear cost relative to the sequence length and is expensive for sequences much longer than 4K (or 16K) tokens. To endow the models the ability to learn ultra-long range dependency, we need to lower this cost. What we describe in this paper is a concrete step forward – based on certain task-specific assumptions which appear to generally hold, we outline a formulation that works and delivers the expected improvements.

**(1) Focus on what we need for a task: VIP-token centric compression (VCC).** We hypothesize/find that in many tasks where Transformers are effective, only a small subset of tokens, which we refer to as VIP-tokens, are relevant to the final output (and accuracy) of a Transformer. If these tokens had been identified somehow, we could preserve this information in its entirety and only incur a moderate loss in performance. Now, *conditioned* on *these* specific VIP-tokens, an aggressive compression on the other *non-VIP-tokens*, can serve to reduce (and often, fully recover) the loss in performance while dramatically decreasing the sequence length. This compression must leverage information regarding the VIP-tokens, with the goal of improving the approximation of the representation of the VIP-tokens. In other words, a high-fidelity approximation of the entire sequence is unnecessary. Once this "selectively compressed" input passes through a Transformer layer, the output sequence is decompressed to the original full sequence allowing the subsequent layers to access the full sequence.

**(2) Specialized data structure for compression/decompression.** A secondary, but important practical issue, is reducing the overhead when compressing/decompressing the input/output sequences internally in the network. Ignoring this problem will impact efficiency. We give a simple but specialized data structure to maintain the hidden states of the intermediate layers, where the compression can be easily accessed from the data structure, and explicit decompression can be avoid by updating the data structure: the sequence is never fully materialized in intermediate layers.

**Practical contributions.** Apart from the algorithmic modules above, we show that despite an aggressive compression of the input sequences, we achieve better/competitive performance on a broad basket of long sequence experiments. Compared to baselines, we get much better runtime/memory efficiency. We show that it is now practical to run **standard** Transformer models on sequences of 128K token lengths, with consistent performance benefits (and **no** complicated architecture changes).

## 2 Preliminaries

We review the Transformer layer, related work on efficient Transformers and define notations/simplifications. **BOLD** uppercase letters denote matrices, **bold** lower case letters denote vectors, and regular lower case letters denote scalars or functions.

**Brief review of the Transformer Model.** Fix $n$ to be the sequence length and let $d$ be the embedding dimension. Define an embedding matrix $\mathbf{X} \in \mathbb{R}^{n \times d}$ which gives the $n$ feature vector inputs for a Transformer layer. The output of this Transformer layer, $\mathbf{X}_{new}$, is defined as

$$\mathbf{X}_{new} = \beta(\alpha(\mathbf{X}) + \mathbf{X}) + \alpha(\mathbf{X}) + \mathbf{X} \tag{1}$$

using $\alpha(\mathbf{X})$ as shorthand for $\alpha(\cdot, \cdot, \cdot)$, which is a multi-head attention (MHA) with $\mathbf{X}$ as input for queries, keys, and values, described shortly. Here, $\beta(\cdot)$ is a feed-forward network (FFN). Layer

norms [1] are omitted to reduce clutter. Let the inputs to $\alpha(\cdot, \cdot, \cdot)$ be $\mathbf{Q}, \mathbf{K}, \mathbf{V} \in \mathbb{R}^{n \times d}$ for queries, keys, and values. MHA is defined as:

$$\alpha(\mathbf{Q}, \mathbf{K}, \mathbf{V}) := \mathrm{cat}_{i=1}^{i=g} \left[ \mathrm{softmax}(\mathbf{Q}\mathbf{W}_{Q,i}\mathbf{W}_{K,i}^\top \mathbf{K}^\top)\mathbf{V}\mathbf{W}_{V,i} \right] \mathbf{W} \tag{2}$$

where $g$ is the number of attention heads, $\{\mathbf{W}_{Q,i}, \mathbf{W}_{K,i}, \mathbf{W}_{V,i}\}$ are trainable projections, and the 'cat' concatenates the outputs of multiple self-attention modules. We omit the biases for notational simplicity. For ease of discussion, let us further simplify the above notation by assuming that $g = 1$, and suppress $\mathbf{W}_{Q,1}, \mathbf{W}_{K,1}, \mathbf{W}_{V,1}, \mathbf{W}$ as well as the normalization in softmax: they will *still* be estimated within the model (i.e., this module remains unchanged) but are tangential to the description of our idea. With these simplifications, the $\alpha(\cdot, \cdot, \cdot)$ can be expressed as:

$$\alpha(\mathbf{Q}, \mathbf{K}, \mathbf{V}) := \exp(\mathbf{Q}\mathbf{K}^\top)\mathbf{V}. \tag{3}$$

Let $\gamma(\cdot)$ be a placeholder for all heavy computations in the Transformer layer above:

$$\gamma(\mathbf{X}) := \beta(\alpha(\mathbf{X}) + \mathbf{X}) + \alpha(\mathbf{X}). \tag{4}$$

We can verify that the output of a Transformer block (parameters are suppressed to reduce clutter) is,

$$\mathbf{X}_{new} = \gamma(\mathbf{X}) + \mathbf{X}. \tag{5}$$

A Transformer model consists of many such layers: the input of each layer is the output $\mathbf{X}_{new}$ from the previous layer. Let $l$ be the number of layers, then the overall complexity is $\mathcal{O}(ln^2d + lnd^2)$.

**Efficient Transformers.** Many efficient self-attention methods are available to reduce the $\mathcal{O}(ln^2d)$ cost. We list a few models noting that this list is not exhaustive. Performer [6], Random Feature Attention [25], and Nyströmformer [35] propose different low rank approximations of the self-attention matrices. Longformer [2] and Big Bird [38] describe global + local sparse attention. Reformer [17] and YOSO [40] exploit locality sensitive hashing for approximating the self-attention matrix. MRA attention [39] gives a multi-resolution approximation of the self-attention matrices. Memorizing Transformers [34] and RMT [3] follow a recurrent design and store the past context in an external memory module. By sequentially processing one segment of input sequences at one time, they avoid blowing up the memory when processing long sequences.

**Efficient Transformers do not scale well to ultra-long sequences**. Existing self-attention mechanisms often reduce the quadratic cost of MHA to linear. But so far, most experiments report sequence lengths of up to 4K, with some exceptions [2, 38, 14]. Beyond 4K, the linear cost (on $n$) for both computing efficient attentions and FFN makes the cost prohibitive, especially for large models. For example, although LongT5 [14] can train on sequence lengths of up to 16K tokens with an efficient self-attention and shows promising results for longer sequences, it is slower and needs a sizable amount of compute (for example, see Fig. 1). Similar to efficient self-attention with linear cost attention, [34] and [3] do not try to reduce the linear cost (on sequence length) for FFN, so the computation might still be expensive for processing ultra long sequences. On the other hand, our method seeks to reduce the overall cost (both self-attention and FFN) of processing ultra long sequences and processes the entire sequence simultaneously.

**Other alternatives for sequence compression?** Compressing input sequences for efficiency reasons in Transformers is not a new idea. For example, [8] and [16] propose pyramid Transformer variants that progressively compress the sequence as the layers grow deeper via pooling or core-set selection. [23] proposes adaptively compressing the sequence based on the predicted semantic boundaries within the sequence. [26] proposes compressing the fine-grained past activations to coarser memories. There are **three** key differences with our approach. First, all methods listed above are *task agnostic*. They seek compressed/smaller representations to represent the *original* sequence well. Our formulation places no emphasis on representing the original sequence, as long as information pertinent to the VIP-tokens is preserved as much as possible. Second, once these methods compress the sequence, the residual information is lost (for the deeper layers or the later time steps). Our entire approach is predicated on avoiding this loss – we maintain access to the full sequence at each layer (via residual connection at least). Lastly, some of these ideas often involve an $n^2$ dependence on the sequence length in the initial stages of their formulation, making long sequence experiments problematic.

## 3 VIP-Token Centric Compression (VCC)

Our main goal is to reduce the dependency on $n$ (but *not* by modifying Transformer internals). To do this, we describe a scheme that compresses the input sequence of a Transformer layer and

decompresses the output sequence, resulting in a model whose complexity is $\mathcal{O}(lrd^2 + lr^2d + lr\log(n_c)d + lrn_pd + nd)$. Here, $r$ is the length of the compressed sequence, $n_p$ is the number of VIP-tokens described shortly, and $n_c$ is the size of non-VIP/remaining tokens. So, we have $n_p + n_c = n$ and assume $n_p \ll r \ll n$. (complexity analysis is provided in the Appendix.)

**Parsing the complexity term:** Let us unpack the term to assess its relevance. The first two terms $\mathcal{O}(lrd^2 + lr^2d)$ pertain to the cost for a Transformer, while the remaining terms are the overhead of compression and decompression. The term $\mathcal{O}(lr\log(n_c)d + lrn_pd)$ is the overhead of compression and updating our data structure at each layer. The $\mathcal{O}(nd)$ term corresponds to pre-processing involving converting the hidden states into our data structure and post-processing to recover the hidden states from the data structure. Note that unlike the dependence on $n$ for vanilla Transformers, this $\mathcal{O}(nd)$ is incurred only at the input/output stage of the Transformer, but **not** at any intermediate layers.

**High level design choices.** We use the *standard* Transformer layers with a *standard* feed-forward network (which results in $d^2$ in the first term) and *standard* quadratic cost self-attention (which gives the $r^2$ factor in the second term). Why? These choices help isolate the effect of incorporating their efficient counterparts. The proposed algorithm operates on the *input/output of each Transformer layer* leaving the Transformer module itself unchanged. Therefore, our goals are distinct from the literature investigating efficient self-attentions and efficient feed-forward networks. This is because one can replace these two vanilla modules with *any other* efficient alternatives to further reduce the $r^2$ and $d^2$ terms directly. Despite these quadratic terms, our approach is faster than baselines (§4).

We will first describe our general idea, as shown in Fig. 2, which uses VIP-tokens to guide the compression/decompression of the input/output of a Transformer layer so that it only needs to process the compressed sequence (§3.1, §3.2). Then, we will discuss an instantiation of the compression process, by adapting a multi-resolution analysis technique (§3.3). Lastly, we will introduce a data structure which allows more efficient compression/decompression (§3.4).

## 3.1 Elevating the Importance of a Few Tokens: VIP-Tokens

Let us start with the simplest compression, which identifies a linear transformation $\mathbf{S} \in \mathbb{R}^{r \times n}$ which acts on the input, resulting in a smaller representation $\mathbf{SX} \in \mathbb{R}^{r \times d}$. Of course, a smaller $r$ implies that more information about $\mathbf{X}$ is lost. But we find that in many tasks, only the embedding representations of *a few* tokens drive the final prediction: we refer to these tokens as *VIP-tokens*.

**Examples of VIP-tokens:** Observe that only the embedding outputs of masked tokens in masked language modeling [10] and the CLS token in sequence classification [10, 11] are/is used for prediction. In question answering, only the questions and possible answers associated with the questions are used for prediction. It is important to note that the masked tokens, CLS tokens, and question tokens are **(1)** defined by the tasks and **(2)** *known* to the model (although the embedding representation of these tokens are unknown). These VIP-tokens can be viewed as a task or question that is given to the model. The model can process the sequence with a specific goal in mind so that the model can skip/skim less relevant segments. Our general principle involves choosing *a set of tokens* as the VIP-tokens that **(1)** are important to the specific task goals and **(2)** easily pre-identifiable by the user.

*Caveats.* Not all important tokens can be pre-identified. For example, the tokens in the correct answer span in answer span prediction are also important to the specific goals, but are difficult to pre-identify, so only the question tokens (and not the answer tokens) are used as VIP-tokens. We assume that any other tokens that are relevant for prediction should have high dependency with these VIP-tokens. For example, the answer tokens should have high dependency (in self-attention) to the question tokens.

**VIP-tokens occupy the front seats.** VIP-tokens can occur anywhere within a sequence. But we can re-order the sequence as well as the positional encodings so that VIP-tokens are always at the *head of sequence* to make analysis/implementation easier. With this layout, let $\mathbf{P} \in \mathbb{R}^{n_p \times d}$ be the VIP-tokens and $\mathbf{C} \in \mathbb{R}^{n_c \times d}$ be the non-VIP/remaining tokens, $\mathbf{X}$ can be expressed as

$$\mathbf{X} = \begin{bmatrix} \mathbf{P} \\ \mathbf{C} \end{bmatrix} \tag{6}$$

This is possible since Transformer is permutation invariant when permuting positional encodings (embeddings or IDs) along with tokens. This re-ordering is performed only once for the input of the Transformer model, then the outputs generated by the model are rearranged to their original positions.

Re-ordering makes the analysis, implementation and presentation of our method much clearer and simpler. In fact, placing VIP tokens at the end of the sequence can also serve the same purpose.

From the above discussion, it is clear that one needs to make sure that after compressing the input tokens $\mathbf{X}$, the VIP-tokens must still stay (more or less) the same, and the compression matrix $\mathbf{S}$ must be *VIP-token dependent*. We hypothesize that such *VIP-token dependent* compression matrices require a much smaller dimension $r$, compared to *VIP-token agnostic* compression matrices.

### 3.2 VIP-Token Centric Compression (VCC): An Initial Proposal

For a Transformer layer, let $\mathbf{X}$ denote its input matrix. Express the output of this layer as follows:

$$\mathbf{X}_{new} = \mathbf{S}^\dagger \gamma(\mathbf{SX}) + \mathbf{X} \tag{7}$$

where $\mathbf{S} \in \mathbb{R}^{r \times n}$ is a *compression* matrix compressing $\mathbf{X}$ to a smaller representation and $\mathbf{S}^\dagger$ is the pseudo inverse for *decompression*. With the layout in (6), we can write (7) as

$$\begin{bmatrix} \mathbf{P}_{new} \\ \mathbf{C}_{new} \end{bmatrix} = \mathbf{S}^\dagger \gamma \left( \mathbf{S} \begin{bmatrix} \mathbf{P} \\ \mathbf{C} \end{bmatrix} \right) + \begin{bmatrix} \mathbf{P} \\ \mathbf{C} \end{bmatrix} \tag{8}$$

where $\mathbf{P}_{new}$ and $\mathbf{C}_{new}$ are the new embeddings for $\mathbf{P}$ and $\mathbf{C}$.

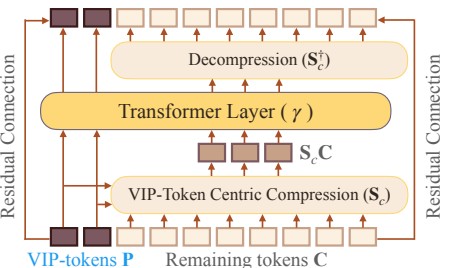

Figure 2: Diagram that illustrates a Transformer layer with VIP-token centric sequence compression.

**Always reserve seats for VIP-tokens.** What is a useful structure of $\mathbf{S}$? Since $\mathbf{P}_{new}$ is the embedding output for the VIP-tokens $\mathbf{P}$, we want them to be fully preserved. To achieve this, we impose the following structure on $\mathbf{S}$ and $\mathbf{S}^\dagger$:

$$\mathbf{S} = \begin{bmatrix} \mathbf{I}_{n_p} & 0 \\ 0 & \mathbf{S}_c \end{bmatrix} \qquad \mathbf{S}^\dagger = \begin{bmatrix} \mathbf{I}_{n_p} & 0 \\ 0 & \mathbf{S}_c^\dagger \end{bmatrix}. \tag{9}$$

The rearrangement simply says that we will avoid compressing $\mathbf{P}$. But rewriting it in this way helps us easily unpack (8) to check the desired functionality of $\mathbf{S}_c$.

**Prioritize information in VIP-tokens.** Our goal is to ensure $\mathbf{P}_{new}$ generated from the compressed sequence in (8) will be similar to its counterpart from the uncompressed sequence. Let us check (8) using the compression matrix $\mathbf{S}$ defined in (9) first. We see that

$$\begin{bmatrix} \mathbf{I}_{n_p \times n_p} & 0 \\ 0 & \mathbf{S}_c^\dagger \end{bmatrix} \gamma \left( \begin{bmatrix} \mathbf{P} \\ \mathbf{S}_c \mathbf{C} \end{bmatrix} \right) = \begin{bmatrix} \beta(\alpha(\mathbf{P}, \mathbf{SX}, \mathbf{SX}) + \mathbf{P}) + \alpha(\mathbf{P}, \mathbf{SX}, \mathbf{SX}) \\ \mathbf{S}_c^\dagger \beta(\alpha(\mathbf{S}_c \mathbf{C}, \mathbf{SX}, \mathbf{SX}) + \mathbf{S}_c \mathbf{C}) + \mathbf{S}_c^\dagger \alpha(\mathbf{S}_c \mathbf{C}, \mathbf{SX}, \mathbf{SX}) \end{bmatrix}. \tag{10}$$

The orange color identifies terms where $\mathbf{P}_{new}$ interacts with other compression-related terms $\mathbf{C}$ and/or $\mathbf{S}_c$. We primarily care about $\mathbf{P}_{new}$ in (8), so the first (orange) row in (10) is the main concern. We see that $\mathbf{P}_{new}$ only depends on the compressed $\mathbf{SX}$ via $\alpha(\mathbf{P}, \mathbf{SX}, \mathbf{SX})$. We can further unpack,

$$\alpha(\mathbf{P}, \mathbf{SX}, \mathbf{SX}) = \exp(\mathbf{PX}^\top \mathbf{S}^\top)\mathbf{SX} = \exp(\mathbf{PP}^\top)\mathbf{P} + \exp(\mathbf{PC}^\top \mathbf{S}_c^\top)\mathbf{S}_c \mathbf{C}. \tag{11}$$

Again, $\alpha(\mathbf{P}, \mathbf{SX}, \mathbf{SX})$ depends on $\mathbf{C}$ and $\mathbf{S}_c$ via the second (orange) term. Normalization in softmax is omitted for simplicity of discussion. This helps us focus on the key term that matters: $\exp(\mathbf{PC}^\top \mathbf{S}_c^\top)\mathbf{S}_c \mathbf{C}$. As long as the following approximation using $\mathbf{S}_c$ is good

$$\exp(\mathbf{PC}^\top \mathbf{S}_c^\top)\mathbf{S}_c \approx \exp(\mathbf{PC}^\top), \tag{12}$$

we will obtain a good approximation of $\mathbf{P}_{new}$. Our remaining task is to outline a scheme of finding a compression matrix $\mathbf{S}_c$ such that this criterion can be assured.

### 3.3 A Specific Instantiation via Multi-Resolution Compression

What should be the mechanics of our compression such that (12) holds? In general, to get $\mathbf{S}_c$, we can use any sensible data driven sketching idea which minimizes the error of (12). Doing so efficiently needs a bit of work; we describe the high level idea below and the low-level details are provided in Appendix.

**High level idea.** Ideally, an efficient scheme for constructing $\mathbf{S}_c$ should operate as follows. If some regions of the sequence $\mathbf{C}$ have a negligible impact on (12) (via the orange terms above), the procedure should compress the regions aggressively. If other regions are identified to have a higher impact on (12) (again due to the orange terms above), the procedure should scan these regions more carefully for a more delicate compression. This suggests that procedurally a coarse-to-fine strategy may work. For example, multi-resolution analysis does help in approximating self-attention matrices in Transformers [39], but the formulation in [39] cannot be easily written in a form similar to (12), making it incompatible with our design. Nonetheless, we derive an analogous form (details in Appendix) that can be represented in a similar form as (12) and gives a strategy for obtaining $\mathbf{S}_c\mathbf{C}$.

Specifically, let us define a compressed representation (via averaging) of the $x$-th $s$-length segment of sequence $\mathbf{C}$: $\mathbf{c}_x^s \in \mathbb{R}^d$

$$\mathbf{c}_x^s := \frac{1}{s} \sum_{sx-s<i\leq sx} [\mathbf{C}]_i \tag{13}$$

where $s \in \{k^0, k^1, k^2, \cdots, n_c\}$ assuming $n_c$ is a power of $k$ and $x \in \{1, 2, \cdots, n_c/s\}$. $[\cdot]_i$ refers to the $i$-th row of the input matrix. We fix the increment ratio $k = 2$ for simplicity of discussion. The $s$ represents the resolution of the approximation: it represents the number of non-VIP token embeddings being averaged into a vector $\mathbf{c}_x^s$. Higher $s$ (e.g., $s = 8$ in $\mathbf{c}_1^8$ in Fig. 3) means lower resolution and heavier compression of the corresponding segment. The $x$ represents the location of the $s$-length segment within the sequence $\mathbf{C}$. In our scheme, we compress the sequence $\mathbf{C}$ and use a set of $\mathbf{c}_x^s$ for some selected $s$'s and $x$'s as the rows of the compressed $\mathbf{S}_c\mathbf{C}$ as seen in Fig. 3. The sequence $\mathbf{C}$ is broken into multiple segments of different lengths, then each segment is compressed into a vector $\mathbf{c}_x^s$.

Procedurally, as shown in Fig. 3, our scheme starts with the heaviest compression and progressively refines certain segments of $\mathbf{C}$ guided by the VIP-tokens $\mathbf{P}$. The scheme starts with the heaviest compression that treats $\mathbf{C}$ as a $n_c$-length segment and compresses it to a single $\mathbf{c}_1^{n_c}$. Then, starting with $s = n_c$ (root node), the procedure (1) computes the averaged attention scores between VIP-tokens $\mathbf{P}$ and $\mathbf{c}_x^s$'s for different $x$'s (averaged over all attention heads and all VIP-tokens; only one $\mathbf{c}_1^{n_c}$ at level $s = n_c$). We note that the attention scores are obtained by extracting attention matrices from MHA module (2) of the current Transformer layer when using $\mathbf{P}$ as queries and $\mathbf{c}_x^s$'s as keys. Then, it (2) splits the $s$-length segments corresponding $\mathbf{c}_x^s$'s with higher averaged attention scores (one segment is split in Fig. 3 but we might split more segments, again only one $\mathbf{c}_1^{n_c}$ at level $s = n_c$) into $(s/2)$-length sub-segments: the corresponding $\mathbf{c}_x^{s/2}$ (13) of each sub-segment is computed for finer representation. Then, at next level for $s = n_c/2$, the same procedure proceeds. This process continues until the sub-segments have length

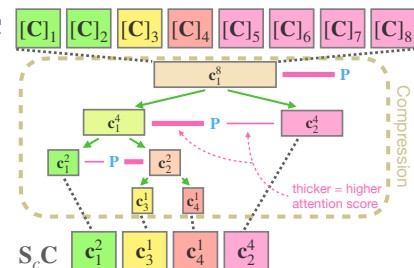

Figure 3: Illustration of multi-resolution compression. $n_c = 8$. Purple line: compute attention scores between $\mathbf{P}$ and different segments. Green arrow: segment with higher attention score is split into two sub-segments. Accordingly, $\mathbf{S}_c\mathbf{C} = \begin{bmatrix} \mathbf{c}_1^2 & \mathbf{c}_3^1 & \mathbf{c}_4^1 & \mathbf{c}_2^4 \end{bmatrix}^\top$ is constructed.

1. We note that this procedure is guided by the VIP-tokens $\mathbf{P}$ and designed to maximally reduce the error of approximating (12). No additional learnable parameters are introduced for this scheme. The technical details of this algorithm are less relevant for our overall approach, but for interested readers, the details are discussed in the Appendix.

**How good is this approximation?** The output $\mathbf{P}_{new}$ (8) is well approximated since the approach preserves the relevant components of $\mathbf{C}$ that have a high impact on the output $\mathbf{P}_{new}$. Further, if the VIP-tokens $\mathbf{P}$ have high attention weights for some rows of $\mathbf{C}$, then the corresponding row in $\mathbf{C}$ will be approximated with higher frequencies (less compressed). So, the output in

Figure 4: Proposed data structure $\mathcal{T}(\mathbf{C})$

$\mathbf{C}_{new}$ (8) for a subset of non-VIP tokens that have a higher dependency with the VIP-tokens will have a better approximation than the others, as desired. This property is useful since some tokens with unknown locations but manifesting a high dependency with the VIP-tokens can also relevant

to the final prediction of a Transformer model in some tasks. The answer in span-based question answering tasks is one example, and our construction ensures that they will be approximated well too.

### 3.4 Efficient Data Structure for Compression/Decompression

By employing the procedure in §3.3 illustrated in Fig. 3, we can find the compressed $\mathbf{S}_c\mathbf{C}$ with an $\mathcal{O}(n_c d + r n_p d)$ cost at each layer. The main cost $\mathcal{O}(n_c d)$ is due to computing $\mathbf{c}_x^s$ defined in (13) for all resolution $s$ and location $x$ by using recursive relation from the bottom up:

$$\mathbf{c}_x^{2s} = \frac{1}{2}\mathbf{c}_{2x-1}^s + \frac{1}{2}\mathbf{c}_{2x}^s \qquad \mathbf{c}_x^1 = [\mathbf{C}]_x \tag{14}$$

We find that these steps could introduce a large overhead. Further, note that if we decompress (apply $\mathbf{S}^\dagger$ to) the output of $\gamma$ for compressed sequence as in (8), the cost is $\mathcal{O}(nd)$ since the number of nonzero entries in $\mathbf{S}^\dagger$ is $n$ (more details in Appendix). As a solution, we now introduce a data structure $\mathcal{T}(\cdot)$ for storing $\mathbf{C}$ and $\mathbf{C}_{new}$, as shown in 4, which enables efficient computation of $\mathbf{c}_x^s$ and eliminates explicit decompression. We note that this data structure is only possible due to the specific structure of $\mathbf{S}_c\mathbf{C}$ constructed in §3.3. Specifically, $\mathcal{T}(\mathbf{C})$ stores $\mathbf{c}_1^{n_c}$ and $\Delta\mathbf{c}_x^s$ defined as

$$\Delta\mathbf{c}_x^s := \mathbf{c}_{\lceil x/2\rceil}^{2s} - \mathbf{c}_x^s \tag{15}$$

for every resolution $s \neq n_c$ and location $x$. Similarly, $\mathcal{T}(\mathbf{C}_{new})$ stores $(\mathbf{c}_{new})_1^{n_c}$ and $\Delta(\mathbf{c}_{new})_x^s$ where $(\mathbf{c}_{new})_x^s$ and $\Delta(\mathbf{c}_{new})_x^s$ are defined similar to (13) and (15) but using $\mathbf{C}_{new}$ instead of $\mathbf{C}$.

Then, given $\mathcal{T}(\mathbf{C})$, any $\mathbf{c}_x^s$ can be retrieved efficiently in $\mathcal{O}(\log(n_c)d)$ cost via recursion:

$$\mathbf{c}_x^s = \mathbf{c}_{\lceil x/2\rceil}^{2s} - \Delta\mathbf{c}_x^s = \mathbf{c}_{\lceil x/4\rceil}^{4s} - \Delta\mathbf{c}_{\lceil x/2\rceil}^{2s} - \Delta\mathbf{c}_x^s = \cdots \tag{16}$$

The only reason we need decompression $\mathbf{S}^\dagger$ is that we need to obtain new representation $\mathbf{C}_{new}$ (no decompression for $\mathbf{P}_{new}$ since $\mathbf{P}$ is uncompressed). Suppose we have $\mathcal{T}(\mathbf{C}_{new})$, then we have an alternative way of getting $\mathbf{C}_{new}$ similar to (16) (note $(\mathbf{c}_{new})_x^1 = [\mathbf{C}_{new}]_x$) without explicit decompression. The key benefit of this data structure is that we can obtain $\mathcal{T}(\mathbf{C}_{new})$ by changing some nodes in $\mathcal{T}(\mathbf{C})$. This only needs updating $\mathcal{O}(r)$ nodes, and each update takes $\mathcal{O}(d)$ cost.

**An example.** We show a $\mathcal{T}(\mathbf{C})$ for $n_c = 8$ in Fig. 4. Let $\mathbf{S}_c\mathbf{C} = \begin{bmatrix} \mathbf{c}_1^2 & \mathbf{c}_3^1 & \mathbf{c}_4^1 & \mathbf{c}_2^4 \end{bmatrix}^\top$ as in Fig. 3. Since the segment $\mathbf{c}_1^2$ is not split into sub-segments $\mathbf{c}_1^1$ and $\mathbf{c}_2^1$, we have (details in Appendix):

$$(\mathbf{c}_{new})_1^1 - \mathbf{c}_1^1 = (\mathbf{c}_{new})_2^1 - \mathbf{c}_2^1 = (\mathbf{c}_{new})_1^2 - \mathbf{c}_1^2 \tag{17}$$

By rearranging (17), we can verify that $\Delta(\mathbf{c}_{new})_1^1, \Delta(\mathbf{c}_{new})_2^1$ in $\mathcal{T}(\mathbf{C}_{new})$ stays the same as $\Delta\mathbf{c}_1^1, \Delta\mathbf{c}_2^1$ in $\mathcal{T}(\mathbf{C})$ and thus do not need to be updated:

$$\begin{aligned} \Delta(\mathbf{c}_{new})_1^1 = (\mathbf{c}_{new})_1^2 - (\mathbf{c}_{new})_1^1 = \mathbf{c}_1^2 - \mathbf{c}_1^1 = \Delta\mathbf{c}_1^1 \\ \Delta(\mathbf{c}_{new})_2^1 = (\mathbf{c}_{new})_1^2 - (\mathbf{c}_{new})_2^1 = \mathbf{c}_1^2 - \mathbf{c}_2^1 = \Delta\mathbf{c}_2^1 \end{aligned} \tag{18}$$

Further, we can verify that only the green nodes in Fig. 4 will be updated. These nodes correspond to the nodes in Fig. 3 that have been traversed. In summary, for each row $\mathbf{c}_x^s$ of $\mathbf{S}_c\mathbf{C}$ (a leaf node in Fig. 3), only the node storing $\Delta(\mathbf{c})_x^s$ and its ancestor nodes in $\mathcal{T}(\mathbf{C})$ must be updated, so the total number of nodes (including their ancestors) being updated is $\mathcal{O}(r)$. Next, we can update the nodes as follows: first, we get representations $(\mathbf{c}_{new})_1^2, (\mathbf{c}_{new})_3^1, (\mathbf{c}_{new})_4^1, (\mathbf{c}_{new})_2^4$ by feeding $\mathbf{S}_c\mathbf{C}$ into Transformer layer (details in Appendix). At level $s = 1$, given $(\mathbf{c}_{new})_3^1$ and $(\mathbf{c}_{new})_4^1$, we **(1)** compute $(\mathbf{c}_{new})_2^2$ via (14), and then **(2)** compute $\Delta(\mathbf{c}_{new})_3^1$ and $\Delta(\mathbf{c}_{new})_4^1$ via (15). The last two values are the new values for $\Delta\mathbf{c}_3^1$ and $\Delta\mathbf{c}_4^1$ in $\mathcal{T}(\mathbf{C})$. At level $s = 2$, given $(\mathbf{c}_{new})_1^2$ and $(\mathbf{c}_{new})_2^2$ computed at previous level, we apply similar procedure to obtain $(\mathbf{c}_{new})_1^4, \Delta(\mathbf{c}_{new})_1^2, \Delta(\mathbf{c}_{new})_2^2$, and the last two values are used to update two nodes in $\mathcal{T}(\mathbf{C})$. It becomes apparent that each node update takes $\mathcal{O}(d)$ cost. Putting it together: the complexity of modifying $\mathcal{T}(\mathbf{C})$ to $\mathcal{T}(\mathbf{C}_{new})$ is $\mathcal{O}(rd)$. The detailed algorithm and complexity analysis are described in Appendix.

By maintaining this data structure, we never need to materialize the entire $\mathbf{C}$ or $\mathbf{C}_{new}$ in any intermediate layer, but instead we use (16) to construct the rows of $\mathbf{S}_c\mathbf{C}$ and perform updates to $\mathcal{T}(\mathbf{C})$ to obtain $\mathbf{C}_{new}$ (represented as $\mathcal{T}(\mathbf{C}_{new})$) at each intermediate layer. At the output of a Transformer, $\mathbf{C}_{new}$ is materialized from $\mathcal{T}(\mathbf{C}_{new})$ at a $\mathcal{O}(n_c d)$ cost via the recursion (16) from the bottom up.

# 4 Experiments

We perform a broad set of experiments to empirically evaluate the performance of our proposed compression. (See hyperparameters/dataset statistics in Appendix.) We evaluate our method on both encoder-only and encoder-decoder architecture types. We compare our method with baselines on a large list of question answering and summarization tasks, where we found long sequences occur most frequently. Then, we study the model performance of scaling to ultra long sequences enabled by our method. Since efficiency is the focus of the efficient baselines and our work, we include runtime efficiency (of a single sequence) in millisecond in each table. (See the procedure for runtime measurement in Appendix.) We also include a discussion on FLOP efficiency in Appendix.

For ease of implementation and hyperparameter selection, we restrict the rows of $\mathbf{S}_c\mathbf{C}$ to have exactly two resolutions for experiments. Specifically, for a pre-defined increment ratio $k$, we split and refine all segments $\mathbf{c}_x^s$ with $s > k$ to $k$-length sub-segments, and select $h$ (pre-defined) $k$-length segments to further split to 1-length sub-segments. So, the rows of $\mathbf{S}_c\mathbf{C}$ would consist of $(n_c/k - h)$ of $\mathbf{c}_x^k$ and $hk$ of $\mathbf{c}_x^1$ for some $x$. To simplify the implementation, we only use the proposed compression in the encoder, and use the vanilla computation in the decoder of encoder-decoder models. We note that our method might be applied to the decoder (more details in Appendix).

Further, we found a few layers of standard Transformer layers to pre-process tokens helps the performance. Therefore, in the initial stage of a Transformer, we segment input sequence into multiple $512$-length segments. For each segment, we use vanilla computation in the first 4 layers (for base models and 6 layers for larger models) of a Transformer. Then, for the remaining layers, segments are concatenated back into one sequence and processed using our proposed compression. There is *no communication* among any segments, so the downstream tasks cannot be solved by these first 4 transformer layers alone, and the initial stage is used just for getting a reasonable representation for the compression to operate on.

**Approximation Quality of VIP-Tokens**. We empirical measured the approximation quality of our VIP centric strategy compared to random strategy (the tree growth in Fig. 3 is not guided by VIP-tokens, but is random) and lazy strategy (each k-length segment is compressed to a token). $\mathbf{P}_{new}$ is the approximated representation of VIP tokens computed with compression and let $\mathbf{P}_{new}^*$ be the ground truth representation of VIP tokens computed without compression. We measure the relative error (defined as $||\mathbf{P}_{new} - \mathbf{P}_{new}^*||_F/||\mathbf{P}_{new}^*||_F$) and correlation coefficient between $\mathbf{P}_{new}$ and $\mathbf{P}_{new}^*$. As shown in Tab. 1, we can verify that the proposed procedure indeed improve the approximation quality of VIP-tokens.

Table 1: Approx quality.

| Compression | Error | Correlation |
|---|---|---|
| Random | 0.403 | 0.919 |
| Lazy | 0.528 | 0.869 |
| VIP centric | 0.137 | 0.991 |

**Encoder-Only Models**. For encoder-only architecture, we compare our method with RoBERTa [22] and three strong baselines: Longformer [2], Big Bird [38], and MRA Attention [39]. We first pretrain a RoBERTa model using masked language modeling task, then for each method, we perform continuous pretraining from the RoBERTa checkpoint to expand the positional embeddings to 4K length and adjust model parameters to adapt approximations used in effi-

Table 2: Dev set results for encoder-only models.

| Method | Size | Length | HotpotQA | | | QuALITY | | WikiHop | |
|---|---|---|---|---|---|---|---|---|---|
| | | | Time | EM | F1 | Time | Accuracy | Time | Accuracy |
| RoBERTa | base | 512 | 19.9 | 35.1 | 44.9 | 21.2 | 39.0 | 19.6 | 67.6 |
| RoBERTa | base | 4K | 422.3 | 62.2 | 76.1 | 403.2 | 39.5 | 414.1 | 75.2 |
| Big Bird | base | 4K | 297.9 | 59.5 | 73.2 | 307.0 | 38.5 | 293.3 | 74.5 |
| Longformer | base | 4K | 371.0 | 59.9 | 73.6 | 368.0 | 27.9 | 369.7 | 74.3 |
| MRA Attention | base | 4K | 203.5 | 63.4 | 77.0 | 200.5 | 38.7 | 199.2 | 76.1 |
| Ours | base | 4K | 114.6 | 60.9 | 74.6 | 126.4 | 39.6 | 108.0 | 75.9 |
| Ours* | base | 4K | 114.6 | 61.4 | 75.0 | 125.7 | 39.5 | 108.0 | 76.1 |
| Ours* | large | 4K | 285.8 | 66.7 | 80.0 | 390.8 | 41.8 | 394.3 | 79.6 |

cient baselines and our method. We verify that our proposed method can be integrated into a pretrained Transformer with some continuous pretraining. But we note that the amount of reduction in log perplexity for our method ($-0.114$) during pre-training is much larger than Longformer ($-0.017$) and Big Bird ($-0.025$) from 50K steps to 250K steps. The continuous pretraining for these baselines might have saturated since only the self-attention is approximated while our method might require more pretraining to adjust the parameters for more aggressive approximation. So, we run a larger scale pretraining for our method; downstream results are in Tab. 2 and Fig. 5, denoted with *. We use HotpotQA [37], QuALITY [24], and WikiHop [33] to assess the language models. HotpotQA is an answer span extraction task, while QuALITY and WikiHop are multi-choice question answering tasks. We set questions and multi-choice answers (for QuALITY and WikiHop) as VIP-tokens.

As shown in Tab. 2, we verify that our method is consistently better compared to Longformer and Big Bird. Our method obtains better accuracy in QuALITY and WikiHop compared to 4K length

Table 3: Dev set results for encoder-decoder models. The left / right values of runtime columns are the runtime for the entire model / the encoder.

| Method | Size | # Param | Length | WikiHop Runtime | EM | F1 | HotpotQA Runtime | EM | F1 | CNN/Dailymail Runtime | R-1 | R-2 | R-L | MediaSum Runtime | R-1 | R-2 | R-L |
|---|---|---|---|---|---|---|---|---|---|---|---|---|---|---|---|---|---|
| T5 | base | 223M | 512 | 25.7 / 20.5 | 66.7 | 69.1 | 26.3 / 20.5 | 34.1 | 44.4 | 40.0 / 20.5 | 43.3 | 20.5 | 40.4 | 39.9 / 20.5 | 30.7 | 14.5 | 28.1 |
| T5 | base | 223M | 4K | 594.3 / 553.7 | 76.2 | 78.1 | 594.3 / 550.6 | 64.2 | 77.5 | 614.4 / 549.4 | 43.8 | 20.9 | 41.0 | 613.5 / 552.9 | 34.9 | 17.2 | 31.9 |
| LongT5 | base | 248M | 4K | 270.7 / 233.9 | 72.7 | 74.8 | 271.3 / 233.7 | 62.3 | 75.7 | 291.6 / 234.9 | 43.3 | 20.6 | 40.5 | 287.3 / 229.5 | 34.9 | 17.3 | 32.0 |
| LED | base | 162M | 4K | 236.6 / 222.9 | 70.0 | 72.4 | 237.4 / 222.9 | 55.1 | 67.9 | 249.4 / 221.8 | 43.3 | 20.0 | 40.5 | - / - | - | - | - |
| Ours | base | 223M | 4K | 181.7 / 148.1 | 76.7 | 78.4 | 155.4 / 127.4 | 64.5 | 77.7 | 195.8 / 139.9 | 43.6 | 20.7 | 40.7 | 196.7 / 140.2 | 34.8 | 17.3 | 31.9 |
| T5 | large | 738M | 512 | 83.5 / 67.0 | 69.1 | 71.4 | 84.1 / 67.0 | 36.9 | 47.8 | 124.6 / 67.0 | 43.8 | 20.7 | 40.9 | 124.5 / 67.0 | 31.9 | 15.5 | 29.1 |
| T5 | large | 738M | 4K | 1738.7 / 1601.0 | 79.1 | 80.7 | 1598.1 / 1598.1 | 68.0 | 81.3 | 1824.8 / 1600.4 | 44.3 | 21.0 | 41.4 | - / - | - | - | - |
| Ours | large | 738M | 4K | 561.4 / 460.5 | 79.0 | 80.6 | 485.3 / 382.8 | 67.8 | 81.0 | 608.1 / 433.8 | 44.4 | 21.4 | 41.5 | 609.7 / 434.4 | 35.8 | 18.2 | 32.8 |
| Ours | 3b | 3B | 4K | 1821.5 / 1441.2 | 80.8 | 82.3 | 1547.7 / 1197.1 | 70.2 | 83.2 | 1930.7 / 1364.8 | 44.8 | 21.5 | 41.9 | 1930.7 / 1364.8 | 36.3 | 18.5 | 33.3 |

| Method | Size | # Param | Length | Qasper Runtime | EM | F1 | QuALITY Runtime | EM | F1 | Arxiv Runtime | R-1 | R-2 | R-L | SummScreenFD Runtime | R-1 | R-2 | R-L |
|---|---|---|---|---|---|---|---|---|---|---|---|---|---|---|---|---|---|
| T5 | base | 223M | 512 | 31.8 / 20.5 | 10.8 | 16.4 | 29.3 / 20.5 | 33.6 | 47.3 | 59.0 / 20.5 | 28.9 | 8.6 | 25.6 | 59.1 / 20.5 | 27.0 | 4.8 | 23.5 |
| T5 | base | 223M | 4K | 608.2 / 551.7 | 13.2 | 29.1 | 596.3 / 551.2 | 34.7 | 47.4 | 645.4 / 549.1 | 44.4 | 18.4 | 39.9 | 647.9 / 551.1 | 31.6 | 6.8 | 27.6 |
| LongT5 | base | 248M | 16K | 1628.5 / 1421.3 | 16.2 | 33.4 | 1633.1 / 1439.7 | 35.8 | 48.5 | 1699.7 / 1370.4 | 48.5 | 21.7 | 43.7 | 1763.4 / 1427.8 | 33.1 | 7.3 | 28.5 |
| LED | base | 162M | 16K | - / - | - | - | - / - | - | - | 1055.8 / 923.6 | 47.8 | 20.6 | 43.2 | - / - | - | - | - |
| Ours | base | 223M | 16K | 538.3 / 391.6 | 16.0 | 30.8 | 557.1 / 419.2 | 36.5 | 48.7 | 672.8 / 392.1 | 48.5 | 21.4 | 43.9 | 670.5 / 390.9 | 33.1 | 7.3 | 28.6 |
| T5 | large | 738M | 512 | 101.9 / 66.4 | 11.3 | 17.0 | 95.8 / 67.1 | 35.3 | 49.0 | 182.2 / 67.1 | 30.5 | 9.1 | 27.1 | 180.9 / 66.5 | 28.3 | 4.9 | 24.9 |
| T5 | large | 738M | 4K | - / - | - | - | 1760.5 / 1596.4 | 37.8 | 50.5 | 1901.5 / 1598.8 | 46.0 | 19.4 | 41.4 | - / - | - | - | - |
| Ours | large | 738M | 16K | 1679.6 / 1120.2 | 16.3 | 33.7 | 1753.6 / 1210.7 | 40.3 | 52.5 | 1959.1 / 1111.0 | 49.5 | 22.2 | 44.7 | 1957.1 / 1109.2 | 34.3 | 7.6 | 29.6 |
| Ours | 3b | 3B | 16K | 6165.4 / 4637.3 | 19.0 | 38.2 | 6398.8 / 4962.7 | 45.2 | 56.0 | 7676.3 / 4642.2 | 49.8 | 22.4 | 45.0 | 7641.5 / 4631.3 | 34.7 | 7.8 | 30.1 |

| Method | Size | # Param | Length | ContractNLI Runtime | EM | F1 | NarrativeQA Runtime | EM | F1 | GovReport Runtime | R-1 | R-2 | R-L | QMSum Runtime | R-1 | R-2 | R-L |
|---|---|---|---|---|---|---|---|---|---|---|---|---|---|---|---|---|---|
| T5 | base | 223M | 512 | 24.0 / 20.5 | 73.5 | 73.5 | 26.8 / 20.5 | 2.0 | 11.3 | 59.1 / 20.5 | 40.5 | 14.8 | 38.2 | 43.5 / 20.5 | 30.2 | 8.0 | 26.5 |
| T5 | base | 223M | 4K | 579.0 / 551.6 | 86.8 | 86.8 | 593.4 / 547.6 | 3.8 | 13.3 | 648.3 / 551.5 | 54.0 | 25.2 | 51.4 | 620.2 / 551.5 | 31.1 | 8.2 | 27.4 |
| LongT5 | base | 248M | 16K | 1564.2 / 1462.5 | 85.1 | 85.1 | 1541.7 / 1370.2 | 5.2 | 15.6 | 1726.4 / 1387.7 | 55.8 | 27.9 | 53.2 | 1721.4 / 1450.7 | 35.7 | 11.7 | 31.4 |
| Ours | base | 223M | 16K | 484.2 / 393.1 | 87.0 | 87.0 | 518.2 / 394.4 | 5.0 | 15.8 | 674.0 / 391.6 | 55.2 | 27.1 | 52.6 | 623.1 / 396.5 | 31.8 | 8.8 | 27.9 |
| T5 | large | 738M | 512 | 78.1 / 67.1 | 74.3 | 74.3 | - / - | - | - | 180.9 / 67.0 | 43.3 | 16.2 | 41.1 | 136.4 / 67.1 | 31.7 | 8.1 | 27.6 |
| T5 | large | 738M | 4K | 1702.4 / 1601.2 | 87.2 | 87.2 | - / - | - | - | - / - | - | - | - | - / - | - | - | - |
| Ours | large | 738M | 16K | 1440.6 / 1122.6 | 87.8 | 87.8 | 1551.7 / 1133.9 | 6.6 | 18.7 | 1955.5 / 1113.8 | 56.3 | 28.0 | 53.8 | 1816.4 / 1134.6 | 34.8 | 10.4 | 30.7 |
| Ours | 3b | 3B | 16K | 5850.2 / 4665.9 | 88.5 | 88.5 | 6055.4 / 4659.4 | 8.2 | 21.2 | 7668.2 / 4642.7 | 56.9 | 28.5 | 54.3 | 7146.7 / 4655.6 | 35.7 | 10.9 | 31.1 |

RoBERTa model, but it is a bit worse than 4k length RoBERTa model on HotpotQA. More pretraining helps close the gap. We also use WikiHop to experiment with method specific hyperparameters (such as block size in Big Bird, window size in Longformer, and compression size $r$ in our method). As shown in Fig. 5, our runtime efficiency frontier is consistently better than the baselines. The key **takeaway** is that our method has a much better runtime efficiency than baselines that have the same sequence length without sacrificing its model performance. Further, we note that our method can be scaled to larger models for accuracy improvement.

**Encoder-Decoder Models**. We compare our method with T5 [27], LongT5 [14], and LED [2]. We use the public pretrained checkpoints for baselines. The pretrained models for our method are obtained by doing continuous pretraining from the public T5 checkpoints using T5's pretraining task [27]. We note that LED-base has 6 encoder/decoder layers compared to 12 encoder/decoder layers in base models of other methods, its accuracy is usually lower. So, LED is evaluated in limited tasks. We use HotpotQA [37], WikiHop [33], CNN/Dailymail [29], MediaSum [43], Arxiv [7], and GovReport [15], SummScreenFD [5], QMSum [42], Nar-

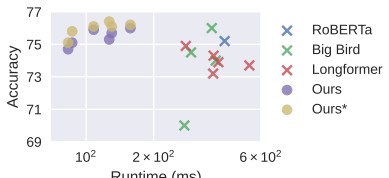

Figure 5: Model runtime vs Wiki-Hop dev accuracy when using different model specific hyperparameters

rativeQA [19], Qasper [9], QuALITY [24], ContractNLI [21] from SCROLLS benchmark [30] to assess the language models. For question answering tasks, we set questions and multi-choice answers (for QuALITY and WikiHop) as VIP-tokens in our method. For query-based summarization, such as QMSum, we use the query as VIP-tokens in our method. For general summartization tasks, we prepend a "summarize:" in each instance and use it as VIP-tokens in our method. Our method achieves matching or better performance in most tasks compared to T5, LongT5, and LED with much higher efficiency (see Tab. 3). Further, the performance monotonically increases with the model size, so our method can scale to larger models.

**Scaling to Longer Sequences**. The prior experiments limit the sequence length to at most 4K or 16K since the baselines can only be scaled up to these lengths. However, our method can be scaled

to much longer sequences. We note that NarrativeQA [19] is an ideal testbed as shown in dataset statistics in Appendix. The results are shown in Tab. 4. The left / middle / right values of runtime column are for the entire model / the encoder / the last 8 layers (out of 12 layers) that uses our compression. The performance monotonically increases as sequence length increases. We note that for sequence length 64K, the performance of model with $k = 64$ is lower than the model with $k = 16$. We suspect that since the results are finetuned from the same model that is pretrained with $k = 16$, the large gap between the two different $k$'s may have a negative impact on finetuning performance. Nevertheless, the performance is still higher than 32K length models.

**Why focus on 4K - 128K lengths?** We believe that the computation required by standard Transformers for processing shorter sequences is not an efficiency bottleneck. As a result, we do not profile the performance of our method for smaller length sequences, since the standard Transformers are sufficiently fast in this case. Further, while our model can be applied to shorter sequences, we suspect that for shorter sequences, there may be less irrelevant information for VIP-tokens. So compressing the irrelevant information will not offer a meaningful speed

Table 4: Dev results of NarrativeQA on base model when scaling sequence length from 16K to 128K.

| Length | Runtime (ms) | $k$ | $h$ | EM | F1 |
|--------|-------------------------|-----|-----|-----|------|
| 16K | 518.2 / 394.4 / 162.4 | 16 | 90 | 5.9 | 16.6 |
| 32K | 946.8 / 671.6 / 212.6 | 32 | 55 | 6.6 | 17.5 |
| 32K | 1027.9 / 751.0 / 298.0 | 16 | 90 | 6.4 | 17.5 |
| 64K | 1848.7 / 1177.2 / 254.8 | 64 | 30 | 7.2 | 18.4 |
| 64K | 2244.8 / 1574.2 / 659.4 | 16 | 90 | 7.5 | 19.3 |
| 128K | 6267.8 / 5125.9 / 1902.2 | 16 | 90 | 8.0 | 19.6 |

up. This is a limitation as the compression works better when there is more compressible information. We have only pushed the sequence lengths to 128K since this length was sufficient to cover a majority of sequence lengths encountered in long sequence tasks (for example, our model is able to process an entire book at once).

## 5 Limitations

Our method assumes that in many tasks, a subset of tokens are disproportionately responsible for the model prediction, the remaining non-VIP-tokens may play a role but are less critical. Our method excels specifically on such tasks by selectively locating relevant information in the sequence for given VIP-tokens. As the experiments show, this choice is effective in many cases but this behavior is not universal. Occasionally, an embedding is pre-computed which must then serve multiple tasks concurrently, e.g., *both* text retrieval and natural language inference. In this case, if we do not know the tasks beforehand, VIP-token selection cannot be meaningfully performed. Further, VIP-token selection requires some understanding of the tasks. However, we believe that a reasonable selection can be made with some generic knowledge for most tasks or use cases. Please see Appendix for VIP-token selection guidelines.

To reduce the complexity of our implementation, the method is currently setup for the encoder module of the Transformer that assumes full access to the entire sequence. The proposed compression might be extended to approximate the computation in the decoder, but it needs more implementation work, so we leave it as future work. Consequently, the current implementation is less useful for decoder-only models (but in the appendix, we discuss some strategies).

## 6 Conclusions

We propose a VIP-token centric sequence compression method to compress/decompress the input/output sequences of Transformer layers thereby reducing the complexity dependency on the sequence length $n$ without sacrificing the model accuracy. Our empirical evaluation shows that our method can be directly incorporated into existing pretrained models with some additional training. Also, it often has much higher efficiency compared to baselines with the same sequence length while offering better or competitive model accuracy. For future work, we believe that extending our method to the decoder of the encoder-decoder and decoder-only models will further boost the efficiency of Transformers while maintaining similar model performance.

**Acknowledgments.** Zeng and Singh were supported in part by funding from the Vilas Board of Trustees and UW–Madison Office of the Vice Chancellor for Research and Graduate Education.

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

# 7 Appendix

## 7.1 Definition of Notations

Table 5: Major notations used

| Notation | Description |
|---|---|
| $l$ | number of layers of a Transformer model |
| $n$ | number of tokens of an input sequence |
| $d$ | model embedding dimension |
| $n_p$ | number of VIP-tokens |
| $n_c$ | number of non-VIP/remaining tokens, so $n_p + n_c = n$ |
| $r$ | length of a compressed sequence |
| $\alpha(\cdot, \cdot, \cdot)$ | multi-head attention taking three inputs for query/key/value embeddings |
| $\beta(\cdot)$ | two-layer feed-forward network |
| $\gamma(\cdot)$ | function representing all heavy computation of a Transformer layer |
| $\mathbf{X}$ | embedding matrix representing a input sequence |
| $\mathbf{P}$ | embedding matrix representing the VIP-tokens |
| $\mathbf{C}$ | embedding matrix representing the non-VIP/remaining tokens |
| $\mathbf{X}_{new}$ | updated embedding matrix of a input sequence, the output of a Transformer layer |
| $\mathbf{P}_{new}$ | updated embedding matrix representing the VIP-tokens |
| $\mathbf{C}_{new}$ | updated embedding matrix representing the non-VIP/remaining tokens |
| $\mathbf{S}$ | compression matrix |
| $\mathbf{S}_c$ | compression submatrix for the non-VIP/remaining tokens |
| $\mathbf{S}^{\dagger}$ | decompression matrix |
| $\mathbf{S}_c^{\dagger}$ | decompression submatrix for the non-VIP/remaining tokens |
| $s$ | resolution of approximation, represents the number of non-VIP token embeddings being averaged |
| $x$ | location of $s$-length segment with the original sequence |
| $k$ | increment ratio of resolution, $s \in \{1, k, k^2, k^3, \cdots, n_c\}$ |
| $h$ | number of $k$-length segments are split into 1-length sub-segments, used for experiments |
| $h_s$ | number of $s$-length segments are split into shorter sub-segments, used in Appendix discussion |
| $\mathcal{J}$ | set of components $\mathbf{b}_x^s$ that is constructed via Alg. 1, the elements are the rows of $\mathbf{S}_c$ |
| $\mathbf{b}_x^s$ | components used for 1-D wavelet transform of scaling $s$ and translation $x$ |
| $\mathbf{c}_x^s$ | local average of $x$-th $s$-length segment of sequence $\mathbf{C}$ |
| $(\mathbf{c}_{new})_x^s$ | local average of $x$-th $s$-length segment of sequence $\mathbf{C}_{new}$ |
| $\mathcal{T}(\cdot)$ | data structure for storing the input sequence $\mathbf{C}$ or $\mathbf{C}_{new}$ |
| $\Delta \mathbf{c}_x^s$ | state stored in $\mathcal{T}(\mathbf{C})$ defined as $\Delta \mathbf{c}_x^s := \mathbf{c}_{\lceil x/2 \rceil}^{2s} - \mathbf{c}_x^s$ |
| $\Delta(\mathbf{c}_{new})_x^s$ | state stored in $\mathcal{T}(\mathbf{C}_{new})$ defined as $\Delta(\mathbf{c}_{new})_x^s := (\mathbf{c}_{new})_{\lceil x/2 \rceil}^{2s} - (\mathbf{c}_{new})_x^s$ |
| $[\cdot]_i$ | $i$-th entry/row of the input vector/matrix |
| $[\cdot]_{i,j}$ | $(i, j)$-th entry of the input matrix |

We provide a table 5 of notations that are used for more than once so that the readers can refer to their definition easily.

## 7.2 Details of Multi-Resolution Compression

We describe the omitted technical details of a modified formulation of [39] to construct $\mathbf{S}_c \mathbf{C}$ and corresponding $\mathbf{S}_c$ satisfying good approximation of

$$\exp(\mathbf{P}\mathbf{C}^{\top}\mathbf{S}_c^{\top})\mathbf{S}_c \approx \exp(\mathbf{P}\mathbf{C}^{\top}). \tag{19}$$

Before diving into the technical details of constructing $\mathbf{S}_c$, we introduce some notations and tools that will be used later. We use $[\cdot]_i$ to refer the $i$-th entry/row of the input vector/matrix and $[\cdot]_{i,j}$ to refer the $(i, j)$-th entry of the input matrix. **BOLD** uppercase letters denote matrices, **bold** lower case letters denote vectors, and regular lower case letters denote scalars. Let $\mathbf{v}_i$ be a vector, when we write a matrix of form

$$\begin{bmatrix} \mathbf{v}_1 & \mathbf{v}_2 & \cdots & \mathbf{v}_m \end{bmatrix}, \tag{20}$$

we treat $\mathbf{v}_i$ as a column of the matrix. When we write a matrix of form

$$\begin{bmatrix} \mathbf{v}_1 \\ \mathbf{v}_2 \\ \cdots \\ \mathbf{v}_m \end{bmatrix}, \tag{21}$$

we treat $\mathbf{v}_i$ as a row of the matrix.

### 7.2.1 Basic Problem Setup

Let $\mathbf{b}_x^s \in \mathbb{R}^{n_c}$ be a multi-resolution component defined as

$$[\mathbf{b}_x^s]_i := \begin{cases} \frac{1}{s} & \text{if } sx - s < i \leq sx \\ 0 & \text{otherwise} \end{cases} \quad (22)$$

for $s \in \{k^0, k^1, k^2, \cdots, n_c\}$ assuming $n_c$ is a power of $k$ (we assume $k = 2$ for simiplicity). Here, $s$ and $x$ represent the scaling and translation (similar to the concepts in wavelet basis) of the component, respectively. Fig. 6 is the visualization of $\mathbf{b}_x^s$.

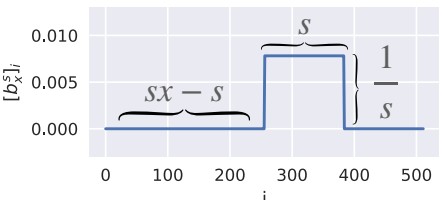

Figure 6: Visualization of $\mathbf{b}_x^s$ for some scaling $s$ and translation $x$. The y axis for different plots are not the same.

Then, any 1D signal $\mathbf{f} \in \mathbb{R}^{n_c}$ can be represented as a linear combination of $\mathbf{b}_x^s$:

$$\mathbf{f} = \sum_{s,x} c_x^s \mathbf{b}_x^s \quad (23)$$

where $c_x^s$ are the coefficients for the linear combination. For a signal with multi-resolution structure (that is, signal has high frequency in some regions and has low frequency in other regions), we can find an approximation $\hat{f}^*$ that can be expressed as a *sparse* linear combination where most coefficients are zeros, as shown in Fig. 7.

$$\mathbf{f} \approx \hat{\mathbf{f}}^* := \sum_{\mathbf{b}_x^s \in \mathcal{J}} c_x^s \mathbf{b}_x^s \quad (24)$$

We denote $\mathcal{J}$ as the set of major components $\mathbf{b}_x^s$ corresponding to the large coefficients, that is, $\mathcal{J} := \{\mathbf{b}_x^s \mid |c_x^s| \text{ being large}\}$. Since the set of all possible $\mathbf{b}_x^s$ is an over-complete dictionary, there are multiple possible linear combinations. To reduce the search space of the best set $\mathcal{J}$, we place a mild restriction on the set $\mathcal{J}$:

$$\left[ \sum_{\mathbf{b}_x^s \in \mathcal{J}} \mathbf{b}_x^s \right]_i \neq 0 \quad \forall i \qquad \langle \mathbf{b}_x^s, \mathbf{b}_{x'}^{s'} \rangle = 0 \quad \forall \mathbf{b}_x^s, \mathbf{b}_{x'}^{s'} \in \mathcal{J}, \mathbf{b}_x^s \neq \mathbf{b}_{x'}^{s'} \quad (25)$$

The conditions state that each entry of signal $\mathbf{f}$ is included in the support region of exactly one component in $\mathcal{J}$. With these tools, we will first describe the approximation when $\mathcal{J}$ is given, then discuss how the approximation connects the set $\mathcal{J}$ to our target $\mathbf{S}_c$ and $\mathbf{S}_c\mathbf{C}$. Finally, we will discuss how to construct this $\mathcal{J}$.

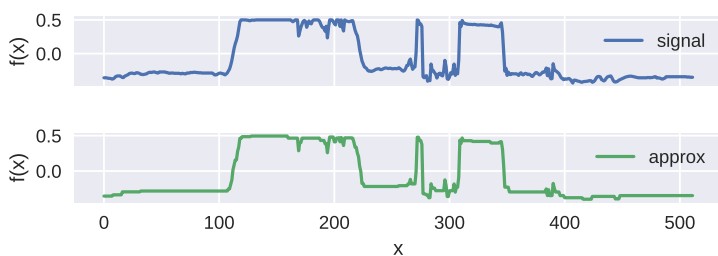

Figure 7: An example of approximating an 1D signal using a truncated wavelet transform with components defined in (22). It uses a set $\mathcal{J}$ of size 79 to represent a signal in $\mathbb{R}^{512}$.

### 7.2.2 Plugging Our Problem into the Setup

A recent result shows that the self-attention matrix $\exp(\mathbf{X}\mathbf{X}^\top)$ has the multi-resolution structure discussed above [39]. Since $\exp(\mathbf{P}\mathbf{C}^\top)$ is a sub-matrix of $\exp(\mathbf{X}\mathbf{X}^\top)$, we conjecture that the multi-resolution structure also holds in $\exp(\mathbf{P}\mathbf{C}^\top)$. As a result, we can find a sparse combination of $\mathbf{b}_x^s$ to represent rows of $\exp(\mathbf{P}\mathbf{C}^\top)$.

**Claim 7.1.** *Given the set $\mathcal{J}$ satisfying restriction* (25)*, we can define an approximation of the $i$-th row of* $\exp(\mathbf{P}\mathbf{C}^\top)$ *similar to* (24) *as illustrated in Fig. 7*

$$\left[ \widehat{\exp(\mathbf{P}\mathbf{C}^\top)}^* \right]_i := \sum_{\mathbf{b}_x^s \in \mathcal{J}} c_x^s \mathbf{b}_x^s \quad (26)$$

*where $c_x^s$ is the optimal solution that minimizes*

$$\left\| \left[\exp(\mathbf{P}\mathbf{C}^\top)\right]_i - \left[\widehat{\exp(\mathbf{P}\mathbf{C}^\top)}^*\right]_i \right\|_2^2 \tag{27}$$

*Then, the approximation can be written as:*

$$\left[\widehat{\exp(\mathbf{P}\mathbf{C}^\top)}^*\right]_{i,j} = \langle \left[\exp(\mathbf{P}\mathbf{C}^\top)\right]_i, \mathbf{b}_x^s \rangle \tag{28}$$

*where $\mathbf{b}_x^s \in \mathcal{J}$ is the component that is supported on $j$ (a.k.a. $[\mathbf{b}_x^s]_j \neq 0$ and there is exactly one $\mathbf{b}_x^s \in \mathcal{J}$ satisfy this condition due to restriction (25)).*

*Proof.* If $\mathcal{J}$ is given, let $\mathbf{B} \in \mathbb{R}^{n_c \times |\mathcal{J}|}$ be a matrix whose columns are elements $\mathbf{b}_x^s \in \mathcal{J}$ and let $\mathbf{c} \in \mathbb{R}^{|\mathcal{J}|}$ be a vector whose entries are the corresponding $c_x^s$:

$$\begin{aligned} \mathbf{B} &:= \begin{bmatrix} \mathbf{b}_{x_1}^{s_1} & \mathbf{b}_{x_2}^{s_2} & \cdots & \mathbf{b}_{x_{|\mathcal{J}|}}^{s_{|\mathcal{J}|}} \end{bmatrix} \\ \mathbf{c} &:= \begin{bmatrix} c_{x_1}^{s_1} & c_{x_2}^{s_2} & \cdots & c_{x_{|\mathcal{J}|}}^{s_{|\mathcal{J}|}} \end{bmatrix}^\top \end{aligned} \tag{29}$$

then the approximation can be expressed as

$$\left[\widehat{\exp(\mathbf{P}\mathbf{C}^\top)}^*\right]_i = \sum_{\mathbf{b}_x^s \in \mathcal{J}} c_x^s \mathbf{b}_x^s = \mathbf{B}\mathbf{c} \tag{30}$$

If we solve for

$$\mathbf{c} := \arg\min_\beta \left\| \left[\exp(\mathbf{P}\mathbf{C}^\top)\right]_i - \mathbf{B}\beta \right\| \tag{31}$$

then

$$\mathbf{c} = (\mathbf{B}^\top \mathbf{B})^{-1} \mathbf{B}^\top \left[\exp(\mathbf{P}\mathbf{C}^\top)\right]_i \tag{32}$$

Due to the restriction (25), the columns of $\mathbf{B}$ are orthogonal, so $\mathbf{B}^\top \mathbf{B}$ is a diagonal matrix:

$$\mathbf{B}^\top \mathbf{B} = \begin{bmatrix} 1/s_1 & & & \\ & 1/s_2 & & \\ & & \ddots & \\ & & & 1/s_{|\mathcal{J}|} \end{bmatrix} \tag{33}$$

We can also write down $\mathbf{B}^\top \left[\exp(\mathbf{P}\mathbf{C}^\top)\right]_i$

$$\mathbf{B}^\top \left[\exp(\mathbf{P}\mathbf{C}^\top)\right]_i = \begin{bmatrix} \langle \left[\exp(\mathbf{P}\mathbf{C}^\top)\right]_i, \mathbf{b}_{x_1}^{s_1} \rangle \\ \langle \left[\exp(\mathbf{P}\mathbf{C}^\top)\right]_i, \mathbf{b}_{x_2}^{s_2} \rangle \\ \cdots \\ \langle \left[\exp(\mathbf{P}\mathbf{C}^\top)\right]_i, \mathbf{b}_{x_{|\mathcal{J}|}}^{s_{|\mathcal{J}|}} \rangle \end{bmatrix} \tag{34}$$

Putting everything together, we have

$$\mathbf{B}\mathbf{c} = \begin{bmatrix} s_1 \mathbf{b}_{x_1}^{s_1} & s_2 \mathbf{b}_{x_2}^{s_2} & \cdots & s_{|\mathcal{J}|} \mathbf{b}_{x_{|\mathcal{J}|}}^{s_{|\mathcal{J}|}} \end{bmatrix} \begin{bmatrix} \langle \left[\exp(\mathbf{P}\mathbf{C}^\top)\right]_i, \mathbf{b}_{x_1}^{s_1} \rangle \\ \langle \left[\exp(\mathbf{P}\mathbf{C}^\top)\right]_i, \mathbf{b}_{x_2}^{s_2} \rangle \\ \cdots \\ \langle \left[\exp(\mathbf{P}\mathbf{C}^\top)\right]_i, \mathbf{b}_{x_{|\mathcal{J}|}}^{s_{|\mathcal{J}|}} \rangle \end{bmatrix} \tag{35}$$

We note that $s\mathbf{b}_x^s$ simply re-scale the entry of $\mathbf{b}_x^s$ such that any non-zero entry becomes 1. Then, let us consider $j$-th entry of $\mathbf{B}\mathbf{c}$. Due to the restriction (25), we have exactly one $\mathbf{b}_x^s \in \mathcal{J}$ whose support region contains $j$, so the $j$-th row of the first matrix at the right hand side of (35) contains exactly a 1 and the remaining entries are 0. Therefore, we have

$$\left[\widehat{\exp(\mathbf{P}\mathbf{C}^\top)}^*\right]_{i,j} = [\mathbf{B}\mathbf{c}]_j = \langle \left[\exp(\mathbf{P}\mathbf{C}^\top)\right]_i, \mathbf{b}_x^s \rangle \tag{36}$$

where $\mathbf{b}_x^s \in \mathcal{J}$ is the component that is supported on $j$, which concludes our proof.

$\square$

### 7.2.3 Efficient Approximation

We note that computing (28) for all $j$ would require access to the entire $\left[\exp(\mathbf{PC}^\top)\right]_i$. We exploit the same strategy as described in [39], so the exponential of inner product is used as an approximation to inner product of exponential.

$$\left[\widehat{\exp(\mathbf{PC}^\top)}\right]_{i,j} := \exp(\langle\left[\mathbf{PC}^\top\right]_i, \mathbf{b}_x^s\rangle) \tag{37}$$

We note that $\langle\left[\mathbf{PC}^\top\right]_i, \mathbf{b}_x^s\rangle$ is the local average of the support region of $\mathbf{b}_x^s$, which is also the $x$-th $s$-length segment of sequence $\left[\mathbf{PC}^\top\right]_i$:

$$\langle[\mathbf{PC}^\top]_i, \mathbf{b}_x^s\rangle = \frac{1}{s}\sum_{[\mathbf{b}_x^s]_j \neq 0}\left[\mathbf{PC}^\top\right]_{i,j} \tag{38}$$

By using some arithmetic manipulations, (37) can be efficiently computed

$$\exp(\langle[\mathbf{PC}^\top]_i, \mathbf{b}_x^s\rangle) = \exp(\frac{1}{s}\sum_{[\mathbf{b}_x^s]_j \neq 0}\left[\mathbf{PC}^\top\right]_{i,j}) = \exp(\frac{1}{s}\sum_{(\mathbf{b}_x^s)_j \neq 0}\langle[\mathbf{P}]_i, [\mathbf{C}]_j\rangle)$$

$$= \exp(\langle[\mathbf{P}]_i, \frac{1}{s}\sum_{[\mathbf{b}_x^s]_j \neq 0}[\mathbf{C}]_j\rangle) = \exp(\langle[\mathbf{P}]_i, \mathbf{c}_x^s\rangle) \tag{39}$$

where $\mathbf{c}_x^s$ is defined in the main text:

$$\mathbf{c}_x^s := \frac{1}{s}\sum_{sx-s<i\leq sx}[\mathbf{C}]_i = \mathbf{b}_x^s\mathbf{C} \tag{40}$$

We note that $\mathbf{c}_x^s$ is the local average of the $x$-th $s$-length segment of sequence $\mathbf{C}$. The $\mathbf{c}_x^s$ can be efficiently computed via

$$\mathbf{c}_x^{2s} = \frac{1}{2}\mathbf{c}_{2x-1}^s + \frac{1}{2}\mathbf{c}_{2x}^s \qquad \mathbf{c}_x^1 = [\mathbf{C}]_x \tag{41}$$

**Claim 7.2.** *Given the set $\mathcal{J}$ satisfying restriction (25), let $\mathbf{S}_c$ be be a matrix whose rows are elements $\mathbf{b}_x^s \in \mathcal{J}$*

$$\mathbf{S}_c = \begin{bmatrix} \mathbf{b}_{x_1}^{s_1} \\ \mathbf{b}_{x_2}^{s_2} \\ \cdots \\ \mathbf{b}_{x_{|\mathcal{J}|}}^{s_{|\mathcal{J}|}} \end{bmatrix} \qquad \mathbf{D} = \begin{bmatrix} s_1 & & & \\ & s_2 & & \\ & & \cdots & \\ & & & s_{|\mathcal{J}|} \end{bmatrix} \tag{42}$$

*Then,*

$$\exp(\mathbf{PC}^\top\mathbf{S}_c^\top)\mathbf{DS}_c = \widehat{\exp(\mathbf{PC}^\top)} \tag{43}$$

*where $\widehat{\exp(\mathbf{PC}^\top)}$ is defined as (37).*

*Proof.* Consider $i$-th row of $\exp(\mathbf{PC}^\top\mathbf{S}_c^\top)$,

$$\begin{aligned}\left[\exp(\mathbf{P}(\mathbf{S}_c\mathbf{C})^\top)\right]_i &= \exp([\mathbf{P}]_i(\mathbf{S}_c\mathbf{C})^\top) \\ &= \exp([\mathbf{P}]_i\begin{bmatrix}\mathbf{c}_{x_1}^{s_1} & \mathbf{c}_{x_2}^{s_2} & \cdots & \mathbf{c}_{x_{|\mathcal{J}|}}^{s_{|\mathcal{J}|}}\end{bmatrix}) \\ &= \begin{bmatrix}\exp(\langle[\mathbf{P}]_i, \mathbf{c}_{x_1}^{s_1}\rangle) & \cdots & \exp(\langle[\mathbf{P}]_i, \mathbf{c}_{x_{|\mathcal{J}|}}^{s_{|\mathcal{J}|}}\rangle)\end{bmatrix}\end{aligned} \tag{44}$$

Then, we have

$$\left[\exp(\mathbf{P}(\mathbf{S}_c\mathbf{C})^\top)\mathbf{DS}_c\right]_i = \begin{bmatrix}\exp(\langle[\mathbf{P}]_i, \mathbf{c}_{x_1}^{s_1}\rangle) & \cdots & \exp(\langle[\mathbf{P}]_i, \mathbf{c}_{x_{|\mathcal{J}|}}^{s_{|\mathcal{J}|}}\rangle)\end{bmatrix}\begin{bmatrix}s_1\mathbf{b}_{x_1}^{s_1} \\ \cdots \\ s_{|\mathcal{J}|}\mathbf{b}_{x_{|\mathcal{J}|}}^{s_{|\mathcal{J}|}}\end{bmatrix} \tag{45}$$

We note that $s\mathbf{b}_x^s$ simply re-scales the entry of $\mathbf{b}_x^s$ such that any non-zero entry becomes 1. Then, let us consider $j$-th entry of $\left[\exp(\mathbf{PC}^\top\mathbf{S}_c^\top)\mathbf{DS}_c\right]_i$. Due to the restriction (25), we have exactly one

$\mathbf{b}_x^s \in \mathcal{J}$ whose support region contains $j$, so the $j$-th column of the second matrix in the right hand side of (45) contains exactly a 1 and the remaining entries are 0. Therefore, we have

$$\left[\exp(\mathbf{P}(\mathbf{S}_c\mathbf{C})^\top)\mathbf{D}\mathbf{S}_c\right]_{i,j} = \exp(\langle[\mathbf{P}]_i\,,\mathbf{c}_x^s\rangle) = \exp(\langle[\mathbf{P}\mathbf{C}^\top]_i, \mathbf{b}_x^s\rangle) = \left[\widehat{\exp(\mathbf{P}\mathbf{C}^\top)}\right]_{i,j} \qquad (46)$$

where $\mathbf{b}_x^s \in \mathcal{J}$ is the component that is supported on $j$. The second equality is based on (39).

$\square$

**Claim 7.3.** *If $\mathbf{S}_c$ and $\mathbf{D}$ are defined as Claim 7.2, the pseudo inverse of $\mathbf{S}_c$ is simply $\mathbf{S}_c^\dagger = \mathbf{S}_c^\top\mathbf{D}$, so each row of $\mathbf{S}_c^\dagger$ and $\mathbf{S}^\dagger$ contain exactly a 1 (so the number of nonzero entries of $\mathbf{S}_c^\dagger$ and $\mathbf{S}^\dagger$ are $n_c$ and $n$ respectively).*

*Proof.* Since each row of $\mathbf{S}_c$ is some $\mathbf{b}_x^s \in \mathcal{J}$, due to the restriction (25), for $i \neq j$,

$$\begin{aligned}
\left[\mathbf{S}_c\mathbf{S}_c^\top\mathbf{D}\right]_{i,i} &= \langle[\mathbf{S}_c]_i\,,[\mathbf{D}\mathbf{S}_c]_i\rangle = \langle\mathbf{b}_{x_i}^{s_i}, s_i\mathbf{b}_{x_i}^{s_i}\rangle = s_i\frac{1}{s_i} = 1 \\
\left[\mathbf{S}_c\mathbf{S}_c^\top\mathbf{D}\right]_{i,j} &= \langle[\mathbf{S}_c]_i\,,[\mathbf{D}\mathbf{S}_c]_j\rangle = \langle\mathbf{b}_{x_i}^{s_i}, s_j\mathbf{b}_{x_i}^{s_j}\rangle = s_j 0 = 0
\end{aligned} \qquad (47)$$

As a result, $\mathbf{S}_c\mathbf{S}_c^\top\mathbf{D} = \mathbf{I}$. Further, $\mathbf{S}_c^\top\mathbf{D}\mathbf{S}_c$ is a symmetric matrix. So, all Moore-Penrose conditions are verified. $\mathbf{S}_c^\dagger = \mathbf{S}_c^\top\mathbf{D}$.

From the restriction (25), we have every column of $\S_c$ contains exactly a non-zero entry. Also,

$$\mathbf{S}_c^\dagger = \mathbf{S}_c^\top\mathbf{D} = \begin{bmatrix} s_1\mathbf{b}_{x_1}^{s_1} & s_2\mathbf{b}_{x_2}^{s_2} & \cdots & s_{|\mathcal{J}|}\mathbf{b}_{x_{|\mathcal{J}|}}^{s_{|\mathcal{J}|}} \end{bmatrix} \qquad (48)$$

Since the non-zero entry of $\mathbf{b}_x^s$ is simply $\frac{1}{s}$ by definition, $s\mathbf{b}_x^s$ simply re-scales the entry of $\mathbf{b}_x^s$ such that any non-zero entry becomes 1. As a result, each row of $\mathbf{S}_c^\dagger$ has exactly a 1. Also, by the relation between $\mathbf{S}$ and $\mathbf{S}_c$:

$$\mathbf{S} = \begin{bmatrix} \mathbf{I}_{n_p\times n_p} & 0 \\ 0 & \mathbf{S}_c \end{bmatrix} \qquad \mathbf{S}^\dagger = \begin{bmatrix} \mathbf{I}_{n_p\times n_p} & 0 \\ 0 & \mathbf{S}_c^\dagger \end{bmatrix} \qquad (49)$$

each row of $\mathbf{S}^\dagger$ has exactly a 1.

$\square$

At the end, the approximation

$$\widehat{\exp(\mathbf{P}\mathbf{C}^\top)} = \exp(\mathbf{P}\mathbf{C}^\top\mathbf{S}_c^\top)\mathbf{D}\mathbf{S}_c \approx \exp(\mathbf{P}\mathbf{C}^\top) \qquad (50)$$

does not look exactly as (19), but we can insert a simple diagonal matrix $\mathbf{D}$ to the formulation (19) and make the whole thing work.

### 7.2.4 How to Construct $\mathcal{J}$ for $\mathbf{S}_c$ and $\mathbf{S}_c\mathbf{C}$?

The derivation so far assume access to $\mathcal{J}$, but in practice, we have no knowledge of $\mathcal{J}$ and need to construct $\mathcal{J}$ that leads to good approximation. With the approximation scheme in place, we can now analyze the approximation error, which will be leveraged later to find a reasonable set of components $\mathcal{J}$. The approximation error of $i$-th row of $\exp(\mathbf{P}\mathbf{C}^\top)$ can be expressed as

$$\begin{aligned}
\mathcal{E}_i &:= \left\|\left[\exp(\mathbf{P}\mathbf{C}^\top)\right]_i - \left[\widehat{\exp(\mathbf{P}\mathbf{C}^\top)}\right]_i\right\|_F^2 \\
&= \sum_{j=1}^{n_c}\left(\left[\exp(\mathbf{P}\mathbf{C}^\top)\right]_{i,j} - \left[\widehat{\exp(\mathbf{P}\mathbf{C}^\top)}\right]_{i,j}\right)^2 \\
&= \sum_{\mathbf{b}_x^s \in \mathcal{J}}\sum_{[\mathbf{b}_x^s]_j \neq 0}\left(\left[\exp(\mathbf{P}\mathbf{C}^\top)\right]_{i,j} - \exp(\langle[\mathbf{P}\mathbf{C}^\top]_i, \mathbf{b}_x^s\rangle)\right)^2 \\
&= \sum_{B_x^s \in \mathcal{J}}\exp(\langle[\mathbf{P}\mathbf{C}^\top]_i, \mathbf{b}_x^s\rangle)\sum_{(B_x^s)_j \neq 0}\left(\exp(\left[\mathbf{P}\mathbf{C}^\top\right]_{i,j} - \langle[\mathbf{P}\mathbf{C}^\top]_i, \mathbf{b}_x^s\rangle) - 1\right)^2 \\
&= \sum_{B_x^s \in \mathcal{J}}\exp(\langle[\mathbf{P}]_i\,,\mathbf{c}_x^s\rangle)\sum_{(B_x^s)_j \neq 0}\left(\exp(\langle[\mathbf{P}]_i\,,[\mathbf{C}]_j\rangle - \langle[\mathbf{P}]_i\,,\mathbf{c}_x^s\rangle) - 1\right)^2 \\
&= \sum_{B_x^s \in \mathcal{J}}\exp(\langle[\mathbf{P}]_i\,,\mathbf{c}_x^s\rangle)\sum_{(B_x^s)_j \neq 0}\left(\exp(\langle[\mathbf{P}]_i\,,[\mathbf{C}]_j - \mathbf{c}_x^s\rangle) - 1\right)^2
\end{aligned} \qquad (51)$$

**Algorithm 1** Constructing $\mathcal{J}$

---

**Input:** VIP-tokens $\mathbf{P}$ and $\mathbf{c}_x^s$ for all $s$ and $x$
**Input:** $h_s$: number of $s$-length segments to refine for each $s \in \{2, 4, \cdots, n_c\}$
Initialize empty $\mathcal{J}$
Compute $\mu_1^{n_c}$ (53) and add $\mathbf{b}_1^{n_c}$ (22) to $\mathcal{J}$ (compute root node)
**for** $s \leftarrow n_c, n_c/2, \cdots, 2$ **do**
    Pop $h_s$ elements $\mathbf{b}_x^s$ with the largest $\mu_x^s$ (53) (select nodes with higher attention scores)
    **for** each $\mathbf{b}_x^s$ **do**
        Compute $\mu_{2x-1}^{s/2}, \mu_{2x}^{s/2}$ (53) and add $\mathbf{b}_{2x-1}^{s/2}, \mathbf{b}_{2x}^{s/2}$ (22) to $\mathcal{J}$ (split selected nodes)
    **end for**
**end for**
**Output:** $\mathcal{J}$

---

The second equality and fourth equality are due to (25) and (39). The approximation error is governed by two components multiply together: attention score between $[\mathbf{P}]_i$ and the local average $\mathbf{c}_x^s$ of the $x$-th $s$-length segment of sequence $\mathbf{C}$ and the inner product of $[\mathbf{P}]_i$ with the amount of deviation of $[\mathbf{C}]_j$ from its local average $\mathbf{c}_x^s$.

When $s = 1$, the deviation is simply zero:

$$\mathbf{c}_x^1 = [\mathbf{C}]_x . \tag{52}$$

It is reasonable to assume that the deviation $[\mathbf{C}]_j - \mathbf{c}_x^s$ is smaller if $s$ is smaller. Therefore, this actually suggests a simple heuristic for selecting $\mathcal{J}$: when $\exp(\langle [\mathbf{P}]_i , \mathbf{c}_x^s \rangle)$ is large, we should approximate the $x$-th $s$-length segment of $\mathbf{C}$ with higher resolution (by splitting the segment to shorter sub-segments and using finer approximation). This heuristic describes the selection criteria for one row of $\exp(\mathbf{P}\mathbf{C}^\top)$, which corresponds to a single VIP-token, for multiple rows of $\exp(\mathbf{P}\mathbf{C}^\top)$ (for multiple VIP-tokens), we simply use

$$\mu_x^s = \sum_{i=1}^{n_p} \exp(\langle [\mathbf{P}]_i , \mathbf{c}_x^s \rangle) \tag{53}$$

as selection criteria since $\mathcal{J}$ is shared by all VIP-tokens.

The construction of $\mathcal{J}$ is described in Alg. 1. This algorithm describes the same procedure as the Figure 3 in the main text. The $\mathbf{b}_x^s$'s in $\mathcal{J}$ are the rows of $\mathbf{S}_c$, and the corresponding $\mathbf{c}_x^s$'s (40) are the rows of $\mathbf{S}_c\mathbf{C}$. The budgets $h_2, h_4, \cdots, h_{n_c}$ required by Alg. 1 is used determine the number of components at each resolution that will be added to $\mathcal{J}$. Specifically, there are $2h_{2s} - h_s$ number of components $\mathbf{b}_x^s$ for $s \neq 1$ based on simple calculations. We can choose budgets such that the final size of $\mathcal{J}$ is $r - n_p$ to make the length of compressed sequence to be $r$.

### 7.2.5 How Good is This Approximation?

At high level, the compression $\mathbf{S}_c$ performs more compression on tokens that are not relevant to the VIP-tokens and less compression to tokens that are important to the VIP-tokens. We will discuss it in more details. Since each row of $S^\dagger$ contain exactly a 1 as stated in Claim 7.3, $S^\dagger$ can commute with $\beta$, so in summary, we can write the approximation of the computation of a Transformer layer as

$$\alpha(\mathbf{P}, \mathbf{SX}, \mathbf{SX}) = \exp(\mathbf{P}\mathbf{P}^\top)\mathbf{P} + \exp(\mathbf{P}\mathbf{C}^\top \mathbf{S}_c^\top)\mathbf{D}\mathbf{S}_c\mathbf{C}$$
$$\mathbf{S}_c^\dagger \alpha(\mathbf{S}_c\mathbf{C}, \mathbf{SX}, \mathbf{SX}) = \mathbf{S}_c^\dagger \exp(\mathbf{S}_c\mathbf{C}\mathbf{P}^\top)\mathbf{P} + \mathbf{S}_c^\dagger \exp(\mathbf{S}_c\mathbf{C}\mathbf{C}^\top \mathbf{S}_c^\top)\mathbf{D}\mathbf{S}_c\mathbf{C}$$
$$\begin{bmatrix} \mathbf{P}_{new} \\ \mathbf{C}_{new} \end{bmatrix} = \begin{bmatrix} \beta(\alpha(\mathbf{P}, \mathbf{SX}, \mathbf{SX}) + \mathbf{P}) + \alpha(\mathbf{P}, \mathbf{SX}, \mathbf{SX}) \\ \beta(\mathbf{S}_c^\dagger \alpha(\mathbf{S}_c\mathbf{C}, \mathbf{SX}, \mathbf{SX}) + \mathbf{S}_c^\dagger \mathbf{S}_c\mathbf{C}) + \mathbf{S}_c^\dagger \alpha(\mathbf{S}_c\mathbf{C}, \mathbf{SX}, \mathbf{SX}) \end{bmatrix} + \begin{bmatrix} \mathbf{P} \\ \mathbf{C} \end{bmatrix} \tag{54}$$

Note that $\mathbf{D}$ is added as discussed in (50).

There are four main approximation components (purple) in (54). Taking the fact that $\mathbf{D}\mathbf{S}_c = (\mathbf{S}_c^\dagger)^\top$, all of these approximations are row or column space multi-resolution approximations governed by $\mathbf{S}_c$ matrix. High attention weight implies higher dependency, and the procedure in Alg. 1 refines regions with large attention weights with higher resolutions. Therefore, the token embedding in $\mathbf{C}$ that have higher dependency to $\mathbf{P}$ are better approximated. The output $\mathbf{P}_{new}$ is well approximated by design since the approximation preserves the higher frequency components of the subset of rows of

$\mathbf{C}$ that has high impact on the output $\mathbf{P}_{new}$. Further, the output in $\mathbf{C}_{new}$ corresponding to the subset of rows of $\mathbf{C}$ that have higher dependency with the VIP-tokens will have better approximation than the remaining rows of $\mathbf{C}$. This property addresses the issue that some tokens with unknown locations are also relevant to the final prediction of a Transformer in some tasks. For example, in question answering tasks, candidate answers are usually expected to have large dependency with question tokens (VIP-tokens), so they are approximated well as well. This approximation property is exactly what we need.

### 7.2.6  Relation to [39] that Inspires Multi-Resolution Compression

Our work and [39] can be viewed as operating at slightly different levels of abstractions. While [39] tries to approximate self-attention computation efficiently, our paper proposes a general framework for performing a VIP-token centric compression on the sequence to efficiently handle extremely long sequences (the self-attention module remains completely unchanged). Our VIP-token centric compression involves a number of steps described in the main text. But one of the key steps involves constructing a compression matrix $\mathbf{S}_c$ which has some desirable properties, namely satisfying (19) which we elaborate further below.

Note that for equation (19), we need a matrix $\mathbf{S}_c$ such that the approximated attention matrix involving $\mathbf{P}$ and $\mathbf{C}$ is similar to the true attention matrix involving $\mathbf{P}$ and $\mathbf{C}$. This is precisely where the general idea of [39] can be used. But the formulation in [39] cannot be applied directly in its original form since it cannot give us $\mathbf{S}_c$. Why? One reason is that the formulation in [39] cannot be written as matrix form similar to equation (19). This may be a reason why [39] has to use custom CUDA kernels in their implementation. Nonetheless, the properties of [39] are useful. So we derive the analogous form but for 1D instead: this 1D case is expressed as applying a matrix (this is the $\mathbf{S}_c$ we are looking for) to the signal $\mathbf{C}$.

One bonus of this modification is that it also removes the need for custom CUDA kernels. At a high level, [39] offers a multi-resolution view of the self-attention matrices, and our modified version is best thought of as a similar multi-resolution view of the sequence itself. But we can also substitute in a different means of obtaining $\mathbf{S}_c$ (which could simply be a sketching matrix). Finally, we note that a naive implementation of the resulting modification still requires a $O(n_c d)$ cost due to the computation of $\mathbf{c}_x^s$ for all possible scaling $s$ and translation $x$. There is a similar cost in [39] (second paragraph in section 4.4 in [39]). The data structure we propose reduces this cost.

### 7.3  Details of Proposed Data Structure

In section, we describe some omitted technical details of the proposed data structure $\mathcal{T}(\cdot)$.

### 7.3.1  Why $(\mathbf{c}_{new})_1^1 - \mathbf{c}_1^1 = (\mathbf{c}_{new})_2^1 - \mathbf{c}_2^1 = (\mathbf{c}_{new})_1^2 - \mathbf{c}_1^2$ if $\mathbf{S}_c\mathbf{C} = \begin{bmatrix} \mathbf{c}_1^2 & \mathbf{c}_3^1 & \mathbf{c}_4^1 & \mathbf{c}_2^4 \end{bmatrix}^\top$?

**Claim 7.4.** *Given the set $\mathcal{J}$ satisfying restriction (25), if $\mathbf{b}_x^s \in \mathcal{J}$, then $(\mathbf{c}_{new})_x^s - \mathbf{c}_x^s = (\mathbf{c}_{new})_{x'}^{s'} - \mathbf{c}_{x'}^{s'}$ for all $\mathbf{b}_{x'}^{s'}$ satisfying the support of $\mathbf{b}_{x'}^{s'}$ is contained in the support of $\mathbf{b}_x^s$ (the $x'$-th $s'$-length segment of $\mathbf{C}$ is a sub-segment of the $x$-th $s$-length segment of $\mathbf{C}$).*

*Proof.* To simplify the notations a bit, without loss of generality, we assume $x = 1$. Then, for $i \leq s$, consider $(\mathbf{c}_{new})_i^1$:

$$\begin{aligned}(\mathbf{c}_{new})_i^1 = [\mathbf{C}_{new}]_i &= \left[\mathbf{S}_c^\dagger \beta(\alpha(\mathbf{S}_c\mathbf{C}, \mathbf{SX}, \mathbf{SX}) + \mathbf{S}_c\mathbf{C}) + \mathbf{S}_c^\dagger \alpha(\mathbf{S}_c\mathbf{C}, \mathbf{SX}, \mathbf{SX})\right]_i + [\mathbf{C}]_i \\ &= \left[\mathbf{S}_c^\dagger\right]_i \beta(\alpha(\mathbf{S}_c\mathbf{C}, \mathbf{SX}, \mathbf{SX}) + \mathbf{S}_c\mathbf{C}) + \left[\mathbf{S}_c^\dagger\right]_i \alpha(\mathbf{S}_c\mathbf{C}, \mathbf{SX}, \mathbf{SX}) + \mathbf{c}_i^1\end{aligned} \tag{55}$$

By Claim 7.3, $\mathbf{S}_c^\dagger = \mathbf{S}_c^\top \mathbf{D}$ and $i$-th row of $\mathbf{S}_c^\dagger$ contains exactly a 1. The column that contains 1 in the $i$-th row of $\mathbf{S}_c^\dagger$ is exactly $s\mathbf{b}_1^s$ since $i$ is contained in the support of exactly one components in $\mathcal{J}$ due to the restriction (25). Denote this column index as $j$, then

$$(\mathbf{c}_{new})_i^1 = [\beta(\alpha(\mathbf{S}_c\mathbf{C}, \mathbf{SX}, \mathbf{SX}) + \mathbf{S}_c\mathbf{C})]_j + [\alpha(\mathbf{S}_c\mathbf{C}, \mathbf{SX}, \mathbf{SX})]_j + \mathbf{c}_i^1 \tag{56}$$

Note that this holds for all $i \leq s$. As a result, for $i, i' \leq s$,

$$(\mathbf{c}_{new})_i^1 - \mathbf{c}_i^1 = [\beta(\alpha(\mathbf{S}_c\mathbf{C}, \mathbf{SX}, \mathbf{SX}) + \mathbf{S}_c\mathbf{C})]_j + [\alpha(\mathbf{S}_c\mathbf{C}, \mathbf{SX}, \mathbf{SX})]_j = (\mathbf{c}_{new})_{i'}^1 - \mathbf{c}_{i'}^1 \tag{57}$$

**Algorithm 2** Computation of one Transformer layer with $\mathcal{T}(\mathbf{C})$

---

**Input:** VIP-tokens $\mathbf{P}$ and data structure $\mathcal{T}(\mathbf{C})$
Use Algo. 1 to construct $\mathcal{J}$ but use (63) to retrieve $\mathbf{c}_x^s$ from $\mathcal{T}(\mathbf{C})$
Construct $\mathbf{S}_c, \mathbf{S}_c\mathbf{C}$ associated with $\mathcal{J}$ using Claim 7.2
Compute
$$\begin{bmatrix} \mathbf{P}_{new} \\ \mathbf{S}_c\mathbf{C}_{new} \end{bmatrix} = \begin{bmatrix} \beta(\alpha(\mathbf{P}, \mathbf{SX}, \mathbf{SX}) + \mathbf{P}) + \alpha(\mathbf{P}, \mathbf{SX}, \mathbf{SX}) + \mathbf{P} \\ \beta(\alpha(\mathbf{S}_c\mathbf{C}, \mathbf{SX}, \mathbf{SX}) + \mathbf{S}_c\mathbf{C}) + \alpha(\mathbf{S}_c\mathbf{C}, \mathbf{SX}, \mathbf{SX}) + \mathbf{S}_c\mathbf{C} \end{bmatrix} \tag{62}$$

Set $\mathcal{T}(\mathbf{C}_{new}) \leftarrow \mathcal{T}(\mathbf{C})$
**for** $s \leftarrow 1, 2, 4, \cdots n_c/2$ **do**
    **for** $\mathbf{b}_x^s \in \mathcal{J}$ **do**
        Locate row location $\mathbf{b}_x^s$ in $\mathbf{S}_c$, refer the index as $j$
        Compute $(\mathbf{c}_{new})_x^s = [\mathbf{S}_c\mathbf{C}_{new}]_j$
        Mark $\Delta(\mathbf{c}_{new})_x^s$ dirty
    **end for**
    **for** dirty $\Delta(\mathbf{c}_{new})_x^s$ **do**
        Compute $(\mathbf{c}_{new})_{\lceil x/2 \rceil}^{2s}$ and update $\Delta(\mathbf{c}_{new})_x^s$
        Mark $\Delta(\mathbf{c}_{new})_{\lceil x/2 \rceil}^{2s}$ dirty
    **end for**
**end for**
Update $(\mathbf{c}_{new})_1^{n_c}$
**Output:** VIP-tokens $\mathbf{P}_{new}$ and data structure $\mathcal{T}(\mathbf{C}_{new})$

---

Then,

$$\begin{aligned}
(\mathbf{c}_{new})_{\lceil i/2 \rceil}^2 - \mathbf{c}_{\lceil i/2 \rceil}^2 &= \frac{1}{2}(\mathbf{c}_{new})_{2\lceil i/2 \rceil-1}^1 + \frac{1}{2}(\mathbf{c}_{new})_{2\lceil i/2 \rceil}^1 - \frac{1}{2}\mathbf{c}_{2\lceil i/2 \rceil-1}^1 - \frac{1}{2}\mathbf{c}_{2\lceil i/2 \rceil}^1 \\
&= \frac{1}{2}((\mathbf{c}_{new})_{2\lceil i/2 \rceil-1}^1 - \mathbf{c}_{2\lceil i/2 \rceil-1}^1) + \frac{1}{2}((\mathbf{c}_{new})_{2\lceil i/2 \rceil}^1 - \mathbf{c}_{2\lceil i/2 \rceil}^1) \\
&= (\mathbf{c}_{new})_{2\lceil i/2 \rceil-1}^1 - \mathbf{c}_{2\lceil i/2 \rceil-1}^1
\end{aligned} \tag{58}$$

The rest follows from induction.

$\square$

### 7.3.2   How do we get $(\mathbf{c}_{new})_1^2, (\mathbf{c}_{new})_3^1, (\mathbf{c}_{new})_4^1, (\mathbf{c}_{new})_2^4$ if $\mathbf{S}_c\mathbf{C} = \begin{bmatrix} \mathbf{c}_1^2 & \mathbf{c}_3^1 & \mathbf{c}_4^1 & \mathbf{c}_2^4 \end{bmatrix}^\top$?

**Claim 7.5.** *We have*

$$\mathbf{S}_c\mathbf{C}_{new} = \beta(\alpha(\mathbf{S}_c\mathbf{C}, \mathbf{SX}, \mathbf{SX}) + \mathbf{S}_c\mathbf{C}) + \alpha(\mathbf{S}_c\mathbf{C}, \mathbf{SX}, \mathbf{SX}) + \mathbf{S}_c\mathbf{C}. \tag{59}$$

*And the updated representation $(\mathbf{c}_{new})_x^s$ of the corresponding $\mathbf{c}_x^s$ (a row of $\mathbf{S}_c\mathbf{C}$) is the corresponding row of $\mathbf{S}_c\mathbf{C}_{new}$.*

*Proof.* By definition,

$$\mathbf{C}_{new} = \mathbf{S}_c^\dagger \beta(\alpha(\mathbf{S}_c\mathbf{C}, \mathbf{SX}, \mathbf{SX}) + \mathbf{S}_c\mathbf{C}) + \mathbf{S}_c^\dagger \alpha(\mathbf{S}_c\mathbf{C}, \mathbf{SX}, \mathbf{SX}) + \mathbf{C} \tag{60}$$

Then,

$$\begin{aligned}
\mathbf{S}_c\mathbf{C}_{new} &= \mathbf{S}_c\mathbf{S}_c^\dagger \beta(\alpha(\mathbf{S}_c\mathbf{C}, \mathbf{SX}, \mathbf{SX}) + \mathbf{S}_c\mathbf{C}) + \mathbf{S}_c\mathbf{S}_c^\dagger \alpha(\mathbf{S}_c\mathbf{C}, \mathbf{SX}, \mathbf{SX}) + \mathbf{S}_c\mathbf{C} \\
&= \beta(\alpha(\mathbf{S}_c\mathbf{C}, \mathbf{SX}, \mathbf{SX}) + \mathbf{S}_c\mathbf{C}) + \alpha(\mathbf{S}_c\mathbf{C}, \mathbf{SX}, \mathbf{SX}) + \mathbf{S}_c\mathbf{C}
\end{aligned} \tag{61}$$

Since the $\mathbf{S}_c$ is the same for $\mathbf{S}_c\mathbf{C}_{new}$ and $\mathbf{S}_c\mathbf{C}$, the second statement follows. $\square$

### 7.3.3   Algorithm for Making $\mathcal{T}(\mathbf{C})$ into $\mathcal{T}(\mathbf{C}_{new})$

In this section, we describe the exact algorithm to update $\mathcal{T}(\mathbf{C})$ into $\mathcal{T}(\mathbf{C}_{new})$. The pseudo code is described in Alg. 2 where $\mathbf{c}_x^s$ is computed via

$$\mathbf{c}_x^s = \mathbf{c}_{\lceil x/2 \rceil}^{2s} - \Delta\mathbf{c}_x^s = \mathbf{c}_{\lceil x/4 \rceil}^{4s} - \Delta\mathbf{c}_{\lceil x/2 \rceil}^{2s} - \Delta\mathbf{c}_x^s = \cdots \tag{63}$$

We use the term "dirty" in Alg. 2 to indicate the node needs to be handled due to node updates. This term is commonly used in computer cache implementations to indicate that the data of a specific location has been updated and needs to be accounted for.

## 7.4 Complexity Analysis

In this section, we will discuss the detailed complexity analysis of our proposed method. The overall complexity of our proposed method is $\mathcal{O}(lrd^2 + lr^2d + lr\log(n_c)d + lrn_pd + nd)$ when using the proposed efficient data structure.

### 7.4.1 Preparing Input Sequence to $\mathcal{T}(\mathbf{C})$: $\mathcal{O}(nd)$

At the first layer, we need to permute the rows of $\mathbf{X}$ into $[\mathbf{P}; \mathbf{C}]$, which takes $\mathcal{O}(nd)$ cost. Then, we process $\mathbf{C}$ into $\mathcal{T}(\mathbf{C})$. This requires **(1)** computing $\mathbf{c}_x^s$ defined in (41). $\mathbf{c}_x^1 = [\mathbf{C}]_x$, so no compute is needed. With all $\mathbf{c}_x^1$ given, computing all $\mathbf{c}_x^2$ takes $\mathcal{O}(n_cd/2)$. With all $\mathbf{c}_x^2$ given, computing all $\mathbf{c}_x^4$ takes $\mathcal{O}(n_cd/4)$... So, the cost is

$$\mathcal{O}(n_cd/2 + n_cd/4 + \cdots + d) = \mathcal{O}(n_cd). \tag{64}$$

Then **(2)** computing $\Delta\mathbf{c}_x^s$ for all $s$ and $x$. Computing each $\Delta\mathbf{c}_x^s$ takes $\mathcal{O}(d)$ when given $\mathbf{c}_x^s$ and $\mathbf{c}_{\lceil x/2 \rceil}^{2s}$. The amount of cost is the same as the number of nodes in the tree $\mathcal{T}(\mathbf{C})$, so the cost is $\mathcal{O}(n_cd)$. Note that $n_c < n$, so the overall complexity of the above operations is $\mathcal{O}(nd)$.

### 7.4.2 Constructing $\mathcal{J}, \mathbf{S}_c, \mathbf{S}_c\mathbf{C}$: $\mathcal{O}(lr\log(n_c)d + lrn_pd)$

We can analyze the complexity of constructing $\mathcal{J}$ using Algo. 1. There is only one possible $\mu_x^{n_c}$. Then for each $s$, there are $2h_s$ number of $\mu_x^{s/2}$ being computed since there are 2 components $\mathbf{b}_x^{s/2}$ for each $\mathbf{b}_{x'}^s$. As a result, we need to compute $\mathcal{O}(1 + \sum_s 2h_s)$ number of $\mu_x^{s/2}$. When $\mathbf{c}_x^{s/2}$ is given, the cost of computing a $\mu_x^{s/2}$ is $\mathcal{O}(n_pd)$, so the overall cost of constructing $\mathcal{J}$ is $\mathcal{O}((1 + \sum_s 2h_s)n_pd)$.

Further, at each $s$, the size of $\mathcal{J}$ is increased by $h_s$ since $h_s$ segments are split into $2h_s$ sub-segments, so the size of $\mathcal{J}$ is $\mathcal{O}(\sum_s h_s)$. Since $\mathbf{S}_c \in \mathbb{R}^{(r-n_p)\times n}$ and $|\mathcal{J}| = r - n_p$ as discussed in §7.2.4, $\mathcal{O}(r - n_p) = \mathcal{O}(\sum_s h_s)$. We use $\mathcal{O}(r)$ for simplicity instead of $\mathcal{O}(r - n_p)$. As a result, the overall cost of constructing $\mathcal{J}$ is $\mathcal{O}(rn_pd)$.

The above cost assumes $\mathbf{c}_x^{s/2}$ is given. If we compute all possible $\mathbf{c}_x^{s/2}$ using (41), the cost will be $\mathcal{O}(n_cd)$ as analyzed in §7.4.1. However, if we employ the proposed data structure, each $\mathbf{c}_x^{s/2}$ can be retrieved in at most $\mathcal{O}(\log(n_c)d)$ by recursively computing (63). Since we need to retrieve $\mathcal{O}(1 + \sum_s 2h_s) = \mathcal{O}(r)$ number of $\mathbf{c}_x^s$, the complexity of computing necessary $\mathbf{c}_x^s$ is $\mathcal{O}(r\log(n_c)d)$.

As a result, the complexity of constructing $\mathcal{J}$ is $\mathcal{O}(rn_pd + r\log(n_c)d)$ at each layer. When summing the cost over all layers, the complexity is $\mathcal{O}(lrn_pd + lr\log(n_c)d)$.

By Claim 7.2, the rows of $\mathbf{S}_c$ and $\mathbf{S}_c\mathbf{C}$ are simply the $\mathbf{b}_x^s \in \mathcal{J}$ and the corresponding $\mathbf{c}_x^s$, which are already computed during the construction of $\mathcal{J}$, so we essentially can get these $\mathbf{S}_c$ and $\mathbf{S}_c\mathbf{C}$ for free.

### 7.4.3 Feeding Compressed Sequence into a Transformer Layer: $\mathcal{O}(lrd^2 + lr^2d)$

At each layer, we need to compute

$$\begin{bmatrix} \mathbf{P}_{new} \\ \mathbf{S}_c\mathbf{C}_{new} \end{bmatrix} = \begin{bmatrix} \beta(\alpha(\mathbf{P}, \mathbf{SX}, \mathbf{SX}) + \mathbf{P}) + \alpha(\mathbf{P}, \mathbf{SX}, \mathbf{SX}) + \mathbf{P} \\ \beta(\alpha(\mathbf{S}_c\mathbf{C}, \mathbf{SX}, \mathbf{SX}) + \mathbf{S}_c\mathbf{C}) + \alpha(\mathbf{S}_c\mathbf{C}, \mathbf{SX}, \mathbf{SX}) + \mathbf{S}_c\mathbf{C} \end{bmatrix} \tag{65}$$

for updating $\mathcal{T}(\mathbf{C})$. This is the part of a Transformer layer that requires heavy computation. It can be verified that the complexity of a Transformer layer is $\mathcal{O}(nd^2 + n^2d)$ for a input sequence of length $n$. Now a compressed sequence of length $r$ is fed into a Transformer layer, the cost is simply $\mathcal{O}(rd^2 + r^2d)$. We note that there is an additional re-scaling to plug $\mathbf{D}$ into $\exp(\mathbf{P}\mathbf{C}^\top\mathbf{S}_c^\top)\mathbf{D}\mathbf{S}_c$ during multi-head attention computation discussed in 50. However, the additional cost of applying $\mathbf{D}$ is $\mathcal{O}(rd)$, which does not change the complexity. When summing the cost of all layers, the overall complexity is $\mathcal{O}(lrd^2 + lr^2d)$.

### 7.4.4 Updating $\mathcal{T}(\mathbf{C})$ into $\mathcal{T}(\mathbf{C}_{new})$: $\mathcal{O}(lrd)$

Once (65) is computed, we need to change $\mathcal{T}(\mathbf{C})$ into $\mathcal{T}(\mathbf{C}_{new})$. The cost of change $\mathcal{T}(\mathbf{C})$ into $\mathcal{T}(\mathbf{C}_{new})$ is $\mathcal{O}(rd)$ as analyzed in the main text. For more specific analysis, let us take a look at the first three iterations:

(1) At the first iteration, there are $\mathcal{O}(2h_2)$ number of $(\mathbf{c}_{new})_x^1$ to be computed at the first inner for loop, and there are $\mathcal{O}(2h_2)$ number of $\Delta(\mathbf{c}_{new})_x^1$ to be updated in the second inner for loop. Additional $\mathcal{O}(h_2)$ number of $\Delta(\mathbf{c}_{new})_{\lceil x/2 \rceil}^2$ are masked dirty.

(2) At the second iteration, there are $\mathcal{O}(2h_4)$ number of $(\mathbf{c}_{new})_x^2$ to be computed at the first inner for loop, and there are $\mathcal{O}(2h_4 + h_2)$ number of $\Delta(\mathbf{c}_{new})_x^2$ to be updated in the second inner for loop. The second term is due to the dirty $\Delta(\mathbf{c}_{new})_{\lceil x/2 \rceil}^2$ from the first iteration. Additional $\mathcal{O}(h_4 + \frac{h_2}{2})$ number of $\Delta(\mathbf{c}_{new})_{\lceil x/2 \rceil}^4$ are masked dirty.

(3) At the third iteration, there are $\mathcal{O}(2h_8)$ number of $(\mathbf{c}_{new})_x^4$ to be computed at the first inner for loop, and there are $\mathcal{O}(2h_8 + h_4 + \frac{h_2}{2})$ number of $\Delta(\mathbf{c}_{new})_x^4$ to be updated in the second inner for loop. The second and third term is due to the dirty $\Delta(\mathbf{c}_{new})_{\lceil x/2 \rceil}^4$ from the second iteration. Additional $\mathcal{O}(h_8 + \frac{h_4}{2} + \frac{h_2}{4})$ number of $\Delta(\mathbf{c}_{new})_{\lceil x/2 \rceil}^8$ are masked dirty.

It becomes apparent that if we sum over the number of computes of $(\mathbf{c}_{new})_x^s$ and updates of $\Delta(\mathbf{c}_{new})_x^s$, the total number is $\mathcal{O}(\sum_s 2h_s + 2\sum_s \sum_{j=1}^{\log(s)} \frac{h_s}{2^j}) = \mathcal{O}(\sum_s h_s + \sum_s h_s) = \mathcal{O}(r)$. Since each compute and update takes $\mathcal{O}(d)$ cost, the overall complexity of changing $\mathcal{T}(\mathbf{C})$ into $\mathcal{T}(\mathbf{C}_{new})$ is $\mathcal{O}(rd)$. When summing the cost of all layers, the overall complexity is $\mathcal{O}(lrd)$.

### 7.4.5 Materializing $C_{new}$ from $\mathcal{T}(\mathbf{C}_{new})$ at the Last Layer: $\mathcal{O}(nd)$

At the output of the last layer, we can (1) compute all $(\mathbf{c}_{new})_x^{n_c/2}$ via (63) at a cost of $\mathcal{O}(2d)$, (2) compute $(\mathbf{c}_{new})_x^{n_c/4}$ via (63) at a cost of $\mathcal{O}(4d)$... until all $(\mathbf{c}_{new})_x^1$ are computed. Then, $[\mathbf{C}_{new}]_x = c_x^1$ is materialized from $\mathcal{T}(\mathbf{C}_{new})$ at a total cost of

$$\mathcal{O}(d + 2d + 4d + \cdots + n_c d) = \mathcal{O}(n_c d). \tag{66}$$

Lastly, undoing the permutation so that $[\mathbf{P}_{new}; \mathbf{C}_{new}]$ are re-ordered to the original positions has a complexity of $\mathcal{O}(nd)$. As a result, the overall complexity is $\mathcal{O}(nd)$.

### 7.4.6 Overall Complexity

In summary, the overall complexity of our method is

$$\mathcal{O}(lrd^2 + lr^2d + lr\log(n_c)d + lrn_pd + nd) \tag{67}$$

### 7.5 Experiments

Table 6: Length statistics of each dataset. The values are the percentiles of number of tokens for the specific tokenizers. For T5 tokenizer, the left value of is for sequence lengths of encoder input, and the right value is for sequence lengths of decoder input.

| Percentile | HotpotQA | RoBERTa QuALITY | WikiHop | WikiHop | HotpotQA | T5 Qasper | QuALITY | ContractNLI |
|---|---|---|---|---|---|---|---|---|
| 75th | 1535 | 7603 | 2204 | 2399 / 6 | 1692 / 6 | 7029 / 29 | 7747 / 17 | 2991 / 4 |
| 95th | 1928 | 8495 | 3861 | 4206 / 9 | 2129 / 10 | 10920 / 71 | 8603 / 28 | 5061 / 4 |

| Percentile | NarrativeQA | CNN/Dailymail | MediaSum | Arxiv | T5 SummScreenFD | GovReport | QMSum | MultiNews |
|---|---|---|---|---|---|---|---|---|
| 75th | 90482 / 10 | 1242 / 87 | 2621 / 29 | 13477 / 364 | 12119 / 188 | 13304 / 811 | 19988 / 110 | 3032 / 379 |
| 95th | 260533 / 18 | 1946 / 130 | 5061 / 64 | 26024 / 759 | 16722 / 330 | 23795 / 983 | 31749 / 162 | 6676 / 468 |

We run all experiments on NVIDIA A100 GPUs. All code is implemented using the standard PyTorch framework. No custom CUDA kernels are needed. As a result, it can be easily deployed to other platforms or ML frameworks. We will publish all code and checkpoints necessary for reproducibility concurrently with the paper publication.

Table 7: Dev set results for encoder-only models fintuning on HotpotQA, QuALITY, and WikiHop.

| Method | Size | Length | HotpotQA | | | QuALITY | | WikiHop | |
| | | | Runtime | EM | F1 | Runtime | Accuracy | Runtime | Accuracy |
|---|---|---|---|---|---|---|---|---|---|
| RoBERTa | base | 512 | 19.9 | 35.1 | 44.9 | 21.2 | 39.0 | 19.6 | 67.6 |
| RoBERTa | base | 4k | 422.3 | 62.2 | 76.1 | 403.2 | 39.5 | 414.1 | 75.2 |
| Big Bird | base | 4k | 297.9 | 59.5 | 73.2 | 307.0 | 38.5 | 293.3 | 74.5 |
| Longformer | base | 4k | 371.0 | 59.9 | 73.6 | 368.0 | 27.9 | 369.7 | 74.3 |
| Ours | base | 4k | 114.6 | 60.9 | 74.6 | 126.4 | 39.6 | 108.0 | 75.9 |
| Ours-150k | base | 4k | 114.6 | 60.7 | 74.1 | 126.4 | 39.4 | 108.0 | 76.1 |

### 7.5.1 Runtime Measurement

The runtimes presented in the experiment section are measured runtimes of a complete training step (including both forward and backward). For each method, we use the largest batch size that can fit into a 80GB A100 and measure the average latency of 10 steps. Then, the average latency is divided by the batch size to get the estimated runtime for a single instance. Via this procedure, we seek to measure the peak efficiency of each method when the GPU is at the highest possible utilization.

### 7.5.2 FLOPs

Some might be interested about the FLOP efficiency of each method, so we provide FLOP profiling results for some experiments. The automatic tools that we used for calculating FLOPs is deepspeed's FlopsProfiler. We note that the FLOP profiling tools do not always work and may throw errors or produce incorrect results when the code contains custom CUDA kernels. The results are summarized in Tab. 8 and Tab. 9. × in some entries means the profiler throws errors and cannot estimate the FLOPs numbers.

Table 8: Dev accuracy and FLOPs for encoder-only models. *: some calculations are NOT captured by profiler, so value is underestimated.

| Method | Size | Length | WikiHop | | |
| | | | Runtime | GFLOPs | Accuracy |
|---|---|---|---|---|---|
| RoBERTa | base | 512 | 19.6 | - | 67.6 |
| RoBERTa | base | 4K | 414.1 | 1317 | 75.2 |
| Big Bird | base | 4K | 293.3 | 790 | 74.5 |
| Longformer | base | 4K | 369.7 | × | 74.3 |
| MRA Attention | base | 4K | 199.2 | 697* | 76.1 |
| Ours | base | 4K | 108.0 | 526 | 75.9 |

Also, the profiler throws errors when we just profile the encoders of encoder-decoder models, so the FLOP numbers in Tab. 9 are for the entire models. As shown in the tables, our method has the lowest FLOPs.

Table 9: Dev results and FLOPs for encoder-decoder models.

| Method | Size | Length | WikiHop | | | |
| | | | Runtime | GFLOPs | EM | F1 |
|---|---|---|---|---|---|---|
| T5 | base | 512 | 25.7 / 20.5 | 120 | 66.7 | 69.1 |
| T5 | base | 4K | 594.3 / 553.7 | 1449 | 76.2 | 78.1 |
| LongT5 | base | 4K | 270.7 / 233.9 | 948 | 72.7 | 74.8 |
| LED | base | 4K | 236.6 / 222.9 | × | 70.0 | 72.4 |
| Ours | base | 4K | 181.7 / 148.1 | 663 | 76.7 | 78.4 |

| Method | Size | Length | ContractNLI | | | |
| | | | Runtime | GFLOPs | EM | F1 |
|---|---|---|---|---|---|---|
| T5 | base | 512 | 24.0 / 20.5 | 112 | 73.5 | 73.5 |
| T5 | base | 4K | 579.0 / 551.6 | 1437 | 86.8 | 86.8 |
| LongT5 | base | 16K | 1564.2 / 1462.5 | 4442 | 85.1 | 85.1 |
| Ours | base | 16K | 484.2 / 393.1 | 1933 | 87.0 | 87.0 |

However, we note that FLOP numbers do not always reflect practical latency reductions. Memory bandwidth and latency (which are not captured by FLOP numbers) also play an important role in the overall latency. For example, sparse matrix multiplication (with unstructured sparsity) usually has a much lower FLOP count than dense matrix multiplication. But, the latency reduction is only possible when sparsity is at least 95% or more (depending on the implementation) since sparse matrix multiplication is a memory bandwidth bounded operator.

### 7.5.3 Pretraining

We use a filtered The Pile dataset [13] for all pretrainings. Since we are using public pretrained tokenizers, we want to enable the distribution of pretraining corpus aligns well with the distribution of corpus used to create the tokenizers. As a result, we use tokens per byte as a proxy for alignment of distributions and filter out PubMed Central, ArXiv, Github, StackExchange, DM Mathematics [28],

Table 10: Dev results for encoder-decoder models on MultiNews.

| Method | Size | # Param | Length | MultiNews | | | |
|---|---|---|---|---|---|---|---|
| | | | | Runtime | R-1 | R-2 | R-L |
| T5 | base | 223M | 512 | 59.2 / 20.5 | 42.5 | 15.3 | 39.0 |
| T5 | base | 223M | 4K | 651.2 / 551.8 | 46.4 | 18.2 | 42.6 |
| LongT5 | base | 248M | 8K | 721.7 / 550.6 | 46.7 | 18.3 | 42.9 |
| LED | base | 162M | 8K | 526.5 / 454.2 | 46.6 | 17.8 | 42.7 |
| Ours | base | 223M | 8K | 377.0 / 224.6 | 46.4 | 18.1 | 42.7 |
| T5 | large | 738M | 512 | 180.8 / 67.0 | 43.4 | 15.6 | 39.8 |
| Ours | large | 738M | 8K | 1140.3 / 651.5 | 48.2 | 19.2 | 44.2 |
| Ours | 3b | 3B | 8K | 4094.5 / 2696.0 | 48.9 | 19.4 | 44.7 |

Ubuntu IRC, EuroParl [20], YoutubeSubtitles, and Enron Emails [18] components, which have tokens per byte greater than $0.3$. Then, the remaining corpus of The Pile dataset is used for pretraining.

For encoder-only models, we pretrain RoBERTa for 750K steps. A batch consists of 8,192 sequences of 512 length. The masking ratio for masked language modeling (MLM) is 15%. Then, 4K length models are continuously pretrained from the RoBERTa checkpoints for 300k steps. The positional embeddings are extended by duplicating the pretrained 512 positional embedding multiple times. For 4K length RoBERTa, Longformer, Big Bird and MRA Attention, the batch size is 64, and the masking ratio is 15%. With 15% masking ratio, there are roughly 616 masked tokens scattering in the sequences. We find that using 616 scattered masked tokens as VIP tokens for 4,096 length sequences might not be indicative for VIP-token centric compression, so we use masking ratio 7.5% and

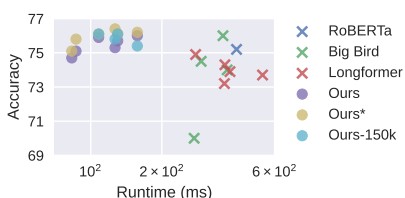

Figure 8: Model runtime vs WikiHop dev accuracy when using different model specific hyperparameters

batch size 128 for our method. The number of masked tokens per sequence is reduced, and the number of total masked token predictions remains the same during pretraining. We note that with larger batch size, the wall clock pretraining runtime for our method is still smaller than baselines. In case that anyone is interested, we also show downstream finetuning on our method pretrained on the same number of tokens but fewer number of masked token predictions in Tab. 7 and Fig. 8, denoted as Ours-150k. The accuracy is consistent with our model pretrained on 300k steps. For the larger scale pretraining denoted with *, we pretrain our method for 250K steps with batch size 512 and masking ratio 7.5%.

For encoder-decoder architecture of our method, we do continuous pretraining from the public checkpoints of T5 for 250K steps with batch size 256 using the masked span prediction. Since each masked span (consists of multiple tokens) is replaced by a single special token, when using masking ratio is 15%, the number of special tokens in a sequence is not too large, we keep masking ratio 15% unchanged.

### 7.5.4 Downstream Finetuning

The statistics of the sequence lengths of instances in each dataset are summarized in Tab. 6. The hyperparameters of all experiments are summarized in Tab 11. When there are multiple values in an entry, it means we perform a hyperparameter search on these values. The amount of search is determined by the size of datasets. If a dataset is relatively large, we only search the learning rate. If a dataset is small, we include batch size and the number of epochs in search. For all tasks, if the sequence lengths are longer than the model length $m$, the sequences will be truncated and only the first $m$ tokens will be used. For encoder-decoder models, we use greedy decoding in sequence generations for simplicity. The maximal decoder output length, specified in Tab. 11, is set such that the maximal length covers the output lengths of more than 99% of instances. When the length of covering 99% of instances is greater than 512, we just set the maximal decoder output length to 512. Additionally, we show one extra experiment on MultiNews [12] in Tab. 10, which is not in the main text due to space limit.

### 7.5.5 Non-Language Tasks

While the focus of this work is on language tasks, the overall design does not limit its application to other tasks as long as the assumption ("a subset of tokens are disproportionately responsible for

Table 11: Hyperparameters for all experiments.

| LM Task | Encoder-Only | | | Encoder-Decoder | | | | |
| | HotpotQA | QuALITY | WikiHop | WikiHop | HotpotQA | CNN/Dailymail | MediaSum | Qasper |
|---|---|---|---|---|---|---|---|---|
| Optimizer | Adam | Adam | Adam | Adam | Adam | Adam | Adam | Adam |
| Weight Decay | 0.01 | 0.01 | 0.01 | 0.01 | 0.01 | 0.01 | 0.01 | 0.01 |
| LR Decay | Linear | Linear | Linear | Linear | Linear | Linear | Linear | Linear |
| Precision | FP16 | FP16 | FP16 | BF16 | BF16 | BF16 | BF16 | BF16 |
| Batch Size | 32 | 16 | 32 | 32 | 32 | 32 | 32 | {16, 32} |
| Learning Rate | {3e-5, 5e-5} | {3e-5, 5e-5} | {3e-5, 5e-5} | {1e-4, 3e-4} | {1e-4, 3e-4} | {1e-4, 3e-4} | {1e-4, 3e-4} | {1e-4, 3e-4} |
| Epochs | 10 | {10, 20} | 10 | 10 | 10 | 10 | 10 | {10, 20} |
| Warmup Steps | 1000 | 200 | 1000 | 1000 | 1000 | 1000 | 1000 | 200 |
| Max Output Length | - | - | - | 32 | 40 | 256 | 256 | 128 |

| LM Task | Encoder-Decoder | | | | | | | |
| | QuALITY | ContractNLI | NarrativeQA | Arxiv | SummScreenFD | GovReport | QMSum | MultiNews |
|---|---|---|---|---|---|---|---|---|
| Optimizer | Adam | Adam | Adam | Adam | Adam | Adam | Adam | Adam |
| Weight Decay | 0.01 | 0.01 | 0.01 | 0.01 | 0.01 | 0.01 | 0.01 | 0.01 |
| LR Decay | Linear | Linear | Linear | Linear | Linear | Linear | Linear | Linear |
| Precision | BF16 | BF16 | BF16 | BF16 | BF16 | BF16 | BF16 | BF16 |
| Batch Size | {16, 32} | {16, 32} | 32 | 32 | {16, 32} | {16, 32} | {16, 32} | 32 |
| Learning Rate | {1e-4, 3e-4} | {1e-4, 3e-4} | {1e-4, 3e-4} | {1e-4, 3e-4} | {1e-4, 3e-4} | {1e-4, 3e-4} | {1e-4, 3e-4} | {1e-4, 3e-4} |
| Epochs | {10, 20} | {10, 20} | 5 | {10, 20} | {10, 20} | {10, 20} | {10, 20} | 10 |
| Warmup Steps | 200 | 1000 | 1000 | 1000 | 200 | 1000 | 100 | 1000 |
| Max Output Length | 90 | 4 | 47 | 512 | 512 | 512 | 310 | 512 |

the model prediction" and VIP-tokens can be reasonably selected) holds or partially holds for the tasks. As a result, we also tested our method on the Long Range Arena (LRA) benchmark [31] and obtained very promising performance among all baselines compared in [39]. We get slightly better performance than the top performing baselines presented in MRA-attention, but note that we are not trying to show that our method outperforms other baselines but to verify that our method can indeed be applied to other tasks.

Note that there are multiple implementations and hyperparameters used in the LRA benchmark, and comparisons across different implementations and hyperparameters is awkward. We use the same implementation and same hyperparameters as [39]. The VIP-token selection of our method in LRA experiments is quite easy. We simply use the prepended CLS token as the only VIP token since it is responsible for the final model classification. The results are shown in Tab. 12. All results except for ours are directly cited from [39]. Since LRA consists of a synthetic task (ListOps), language tasks (Text, Retrieval), and vision tasks (Image, Pathfinder), these results can serve as preliminary evidence indicating the potential applications of our method on other non-language tasks.

Table 12: Test set accuracy of LRA tasks.

| Method | Listops | Text | Retrieval | Image | Pathfinder | Avg |
|---|---|---|---|---|---|---|
| Transformer | 37.1±0.4 | 65.2±0.6 | 79.6±1.7 | 38.5±0.7 | 72.8±1.1 | 58.7±0.3 |
| Performer | 36.7±0.2 | 65.2±0.9 | 79.5±1.4 | 38.6±0.7 | 71.4±0.7 | 58.3±0.1 |
| Linformer | 37.4±0.3 | 57.0±1.1 | 78.4±0.1 | 38.1±0.3 | 67.2±0.1 | 55.6±0.3 |
| SOFT | 36.3±1.4 | 65.2±0.2 | 83.3±1.0 | 35.3±1.3 | 67.7±1.1 | 57.5±0.5 |
| SOFT + Conv | 37.1±0.4 | 65.2±0.4 | 82.9±0.0 | 37.1±4.7 | 68.1±0.4 | 58.1±0.9 |
| Nystromformer | 24.7±17.5 | 65.7±0.1 | 80.2±0.3 | 38.8±2.9 | 73.1±0.1 | 56.5±2.8 |
| Nystrom + Conv | 30.6±8.9 | 65.7±0.2 | 78.9±1.2 | 43.2±3.4 | 69.1±1.0 | 57.5±1.5 |
| YOSO | 37.0±0.3 | 63.1±0.2 | 78.3±0.7 | 40.8±0.8 | 72.9±0.6 | 58.4±0.3 |
| YOSO + Conv | 37.2±0.5 | 64.9±1.2 | 78.5±0.9 | 44.6±0.7 | 69.5±3.5 | 59.0±1.1 |
| Reformer | 18.9±2.4 | 64.9±4.4 | 78.2±1.6 | 42.4±0.4 | 68.9±1.1 | 54.7±0.2 |
| Longformer | 37.2±0.3 | 64.1±0.1 | 79.7±1.1 | 42.6±0.1 | 70.7±0.8 | 58.9±0.1 |
| Big Bird | 37.4±0.3 | 64.3±1.1 | 79.9±0.1 | 40.9±1.1 | 72.6±0.7 | 59.0±0.3 |
| H-Transformer-1D | 30.4±8.8 | 66.0±0.2 | 80.1±0.4 | 42.1±0.8 | 70.7±0.1 | 57.8±1.8 |
| Scatterbrain | 37.5±0.1 | 64.4±0.3 | 79.6±0.1 | 38.0±0.9 | 54.8±7.8 | 54.9±1.4 |
| MRA-2 | 37.2±0.3 | 65.4±0.1 | 79.6±0.6 | 39.5±0.9 | 73.6±0.4 | 59.0±0.3 |
| MRA-2-s | 37.4±0.5 | 64.3±0.8 | 80.3±0.1 | 41.1±0.4 | 73.8±0.6 | 59.4±0.2 |
| Ours | 37.3±0.5 | 65.3±0.2 | 80.9±0.5 | 41.3±1.7 | 74.6±1.5 | 59.9±0.9 |

## 7.6 Potential Application to Decoders

The goal of this work is to reduce the cost of attention **as well as the feedforward network (FFN) block without changing the internals of the transformer block** when dealing with ultra long sequences (which differentiates our work from the few recent ideas that deal with ultra long sequences). Based on the best feasibility of this goal, the scope of application is focused on the encoder and encoder-decoder settings. For standard auto-regressive decoder settings, it might be

difficult to compress the token that is being generated with neighboring tokens. It is in fact challenging to reduce **the cost of FFN** since every generated token will most likely need a full calculation of FFN when it is being generated.

Having said that, there are still clear opportunities for decoder-only models, which are left as our future work. We briefly describe three possible options to do so. **(1)** We can use the input tokens of the decoder as VIP-tokens to compress the representations of context sequence generated by the encoder before Cross Attention computation to reduce the cost of Cross Attention. **(2)** Auto-regressive decoding operates using Causal Attention at each step. This Causal Attention operation requires memory and computation that is linear in the length of the prefix. We can keep the same Causal Attention VIP-token (the representation of the token currently being generated) and apply our method to compress the representations of the previously generated tokens. This reduces the linear complexity of the Causal Attention operation to sublinear. This is useful for reducing the cost of inference. For training, we can break the sequence into two segments: prefix segment and decoding segment. Then, we can use the proposed compression in prefix segment and vanilla computation in decoding segment. To prevent look ahead to the future tokens, we might only use the first token in the decoding segment as VIP-token. **(3)** In many cases, the current large language models (LLMs) are not used directly to generate an ultra long text. Rather, the requirements for processing ultra longer context manifests due to the need to incorporate user input and previous LLM responses (like ChatGPT) or to incorporate search results (such as New Bing). In these cases, there is a prefix context, and LLMs will generate text based on the user prompt and prefix context. We note that our method can indeed be applied to compress the prefix context, and the user prompt and currently generated tokens will be the VIP-tokens.

## 7.7 Practical Questions

**How is the T5's relative positional encoding handled in our method?**

To compute the relative positional bias for self-attention in T5, the position indices of queries and keys are needed (to calculate the position distance between each query and key). After applying our method, the input to the T5 block is the compressed sequence (some tokens are compressed while some tokens remain uncompressed). For each uncompressed token, the position index remains unchanged as its true position in the original sequence. For each new token that represents a compressed segment, the position index is the floored average of position indices of tokens in the segment. In this way, we can minimize the amount of modification needed to the internals of the T5 block, and the relative attention bias for the compressed sequence is an approximation of the attention bias for the original sequence.

**VIP-token selection is task dependent, how to do this selection?**

VIP-token selection requires some understanding of the tasks. However, we believe that a reasonable selection can be made with some generic knowledge for most tasks or use cases. For example, for question answering tasks, we just use questions as VIP-tokens. For classification tasks (for example, Long Range Arane benchmark shown in Appendix), we simply use CLS token as the VIP-token since only CLS token is used for final prediction. For masked language modeling, we use the masked tokens as the VIP-tokens since only these masked tokens are used for final prediction. For question answering, the question tokens (and candidate tokens for multi-choice QA) are used as the VIP-tokens since "they **(1)** are important to the specific task goals and **(2)** easily pre-identifiable by the user." In the worst case, if there is no obvious token to be selected, we can prepend some learnable "latent" tokens or certain user commands (such as "summarize" for summarization tasks as we used in our experiments) and use them as VIP-tokens (in fact, CLS tokens can be thought of as these learnable tokens).

**Why is the performance of our method is better than standard models?**

Our method is an approximation of the standard models, which should be inferior to the standard models, but in some cases, the performance of our method is better than standard models. We believe the reason is that the correct inductive bias improves the performance for tasks with limited amounts of data. Our approach is forced to compress irrelevant information and the attention is carried out on the compressed sequences, but in standard model with standard attention, each token has access to the entire sequence, which enables a larger degree of freedom. As a result, more training data might be required for the model to learn the correct pattern or bias.

