# OpenReview forum: "VCC: Scaling Transformers to 128K Tokens or More by Prioritizing Important Tokens"
_NeurIPS.cc/2023/Conference — NeurIPS 2023 poster_

### Official Review · Reviewer_xiLt · 2023-07-02

**Soundness:** 3 good
**Presentation:** 4 excellent
**Contribution:** 3 good
**Rating:** 6
**Confidence:** 4

**Summary:**

This paper aims at designing efficient Transformers for ultra long sequences. The design is motivated from the intuition that only a small subset of special tokens, which are referred to as VIP tokens, are most relevant to the final prediction. Based on the intuition, a multi-resolution compression is proposed to reduce the computational cost. Experiments are conducted to show the efficiency and accuracy of the proposed model.

**Strengths:**

+ Nice presentation.

+ The observation that only a small subset of special tokens are most relevant to the final prediction is insightful. The proposed task-driven multi-resolution compression technique is well-motivated and technically sound.

+ The experiments are extensive and show reasonable performance gain.

**Weaknesses:**

+ The major concern for the proposed approach is that it **only applies to Transformer encoders**. Many important LLMs today are decoder-only, e.g., GPT, PaLM, LLaMA, etc. The authors formulate their method in an encoder layer (Sections 2 & 3), and show their method can lead to decent performance gains on encoder-only models and encoder-decoder models. However it seems non-trivial to generalize the approach to decoder-only models.

+ The compression also seems to **impede the use of relative positional encoding**. Although there are experiments on T5-based models, no details are provided on how the relative positional information is handled.

+ As discussed at the beginning of Section 3.3, the multi-resolution compression technique is proposed to minimize the approximation error of Eq. (12). However, **neither theoretical guarantees nor numerical simulations on the approximation error are provided**. There seems to be some theoretical results in the appendix but I can't find any analysis on the approximation error (defined in Eq. (33)). Since the multi-resolution compression is the main technical contribution of this paper, I encourage the authors to provide some theoretical analysis and/or numerical experiments on the approximation error in the main body of the paper.

+ **Minor issues**.
   + Line 121: $n_p$ is the size of the VIP-tokens: "size" seems ambiguous here and can be misunderstood as "dimensionality". I suggest writing _$n_p$ is the number of the VIP-tokens_.
   + Eq. (9): $\mathbf I_{n_p\times n_p}$ can be simplified to $\mathbf I_{n_p}$.
   + Line 223, "$s\in\cdots$" should be "_where_ $s\in\cdots$".

**Questions:**

+ Is the proposed approach generalizable to decoder-only models?

+ Is the proposed approach compatible with relative positional encoding? How is the T5's positional encoding handled in the experiments?

+ Can you provide more illustrations on the approximation error of the compression technique?


**Limitations:**

Limitations are discussed in Section 4. However, I recommend the authors to explicitly discuss the applicability to decoder-only models, if the proposed approach only applies to Transformer encoders.

---

> ### Author Rebuttal · Authors · 2023-08-09
>
> Please refer to the global response for our response to the questions about decoder models.
>
> 1. How is the T5's relative positional encoding handled in the experiments?
>
> This detail is not included in the paper, but we will definitely include it in our updated Appendix.
>
> To compute the relative positional bias for self-attention in T5, the position indices of queries and keys are needed (to calculate the position distance between each query and key). After applying our method, the input to the T5 block is the compressed sequence (some tokens are compressed while some tokens remain uncompressed). For each uncompressed token, the position index remains unchanged as its true position in the original sequence. For each new token that represents a compressed segment, the position index is the floored average of position indices of tokens in the segment. In this way, we can minimize the amount of modification needed to the internals of the T5 block, and the relative attention bias for the compressed sequence is an approximation of the attention bias for the original sequence.
>
> 2. Neither theoretical guarantees nor numerical simulations on the approximation error are provided. Can you provide more illustrations on the approximation error of the compression technique?
>
> Thanks for the suggestion. The theoretical guarantee of approximation quality will be a special case of Proposition 4.5 in MRA-attention [1]. We assumed this would be less interesting to the reader but are happy to include it in the paper.
>
> Based on the reviewer’s question, we empirical measured the approximation quality of our VIP centric compression compared to random compression (the tree node splittings in Figure 3 are not guided by VIP-tokens, but are randomly selected) and full compression (no further tree node splittings in Figure 3 after the leaf nodes are all representing k-length segments). Let $\hat{y}$ is the approximated representation of VIP tokens computed with compression and $y$ is the ground truth representation of VIP tokens computed without compression. We measure the relative approximation error (defined as $|| \hat{y} - y ||_F / || y ||_F$ ) (closer to 0 is better) and correlation coefficient between $\hat{y}$ and $y$ (closer to 1 is better). Here is the results:
>
> VIP centric compression: error is $0.1371$ and correlation is $0.9906$
>
> Random compression: error is $0.4025$ and correlation is $0.9187$
>
> Full compression: error is $0.5280$ and correlation is $0.8689$
>
> We will include this result in the main text.
>
> [1] Multi Resolution Analysis (MRA) for Approximate Self-Attention
>
> 3. Minor issues.
>
> Thanks for the suggestions, we will update our text accordingly.
>
> We are grateful for your support and will be glad to answer any additional questions.

---

> > ### Comment · Reviewer_xiLt · 2023-08-10
> > **Thanks for the rebuttal**
> >
> > Thanks for the rebuttal. My concerns are adequately addressed. Thus, I maintain my rating.

---

> > > ### Author Response · Authors · 2023-08-11
> > >
> > > Thank you, we appreciate your time and constructive feedback on the paper.

---

### Official Review · Reviewer_z4Ur · 2023-07-04

**Soundness:** 3 good
**Presentation:** 4 excellent
**Contribution:** 3 good
**Rating:** 6
**Confidence:** 4

**Summary:**

This paper proposes VIP-token centric compression (VCC) as a method to improve the efficiency of Transformers for ultra long sequences. Based on the impact on approximating the representation of the VIP-tokens, VCC can compress the long sequence into a much smaller representation. Thorough experiments validated the effectiveness of the proposed method.

**Strengths:**

* Long sequences input is a practical problem for current LLM-based applications. And the efficient technology for long sequences that this paper focused on has a high application value, in my opinion.
* The paper is well-written, the method is generally novel and the effectiveness is validated by thorough experiments.
* The authors plan to release the code and checkpoints.


**Weaknesses:**

* FLOPs is missing. Runtime is a hardware-dependent metric, and it would be better to see also hardware-independent metrics FLOPs.
* Why only experiments on encoder-only and encoder-decoder architectures? It would be better to see experiments or discussions on recently more popular decoder-only architectures.
* KV Cache is a useful technique for accelerating Transformer inference and has been already integrated into widely used packages such as huggingface transformer and timm. It would be better to see the discussion on whether KV Cache and VCC are orthogonal and whether they can be used simultaneously.
* Missing reference. It would be better to have a discussion on the difference with the previous [1] and concurrent [2] work that scales Transformer to 262k and one million, respectively.

1. Memorizing Transformers. In ICLR 2022.
2. Scaling Transformer to 1M tokens and beyond with RMT. In arXiv.


**Questions:**

See Weaknesses.

**Limitations:**

Adequately addressed.

---

> ### Author Rebuttal · Authors · 2023-08-09
>
> Please refer to the global response for our response to the questions about decoder models.
>
> 1. FLOPs is missing. Runtime is a hardware-dependent metric, and it would be better to see also hardware-independent metrics FLOPs.
>
> Thanks for this nice suggestion. We will include FLOPs for as many experiments as possible (we are not able to profile all methods reliably since some implementations contain custom CUDA kernels). In this case, automatic tools for calculating FLOPs (such as deepspeed’s FlopsProfiler) do not always work and may throw errors or produce incorrect results when the code contains custom CUDA kernels. As a preview, here we provide some FLOP profiling results for the WikiHop downstream task using encoder-only models (last column of table 1 where input sequence length is 4K),
>
> RoBERTa: 1317 GFLOPs
>
> BigBird: 790 GFLOPs
>
> Longformer: FlopsProfiler throw errors
>
> MRA Attention: 697 GFLOPs + Unknown FLOPs for approximating attention that involves custom CUDA kernels, which are NOT captured by FlopsProfiler
>
> VCC: 526 GFLOPs
>
> Our method still has the lowest FLOPs.
>
> However, we note that FLOP numbers do not always reflect practical latency reductions. Memory bandwidth and latency (which are not captured by FLOP numbers) also play an important role in the overall latency. For example, sparse matrix multiplication (with unstructured sparsity) usually has a much lower FLOP count than dense matrix multiplication. But, the latency reduction is only possible when sparsity is at least 95% or more (depending on the implementation) since sparse matrix multiplication is a memory bandwidth bounded operator.
>
> On the other hand, yes, even though it is a hardware-dependent metric, a good runtime requires all contributing factors including FLOPs, memory bandwidth, and memory latency to be reasonably good. This is the reason we reported runtime as our efficiency metric. But we appreciate the suggestion and will definitely include FLOPs in the main paper.
>
> 2. It would be better to see the discussion on whether KV Cache and VCC are orthogonal and whether they can be used simultaneously.
>
> The KV cache for inference is used in the autoregressive setting for the Transformer decoder. The current implementation of our method supports encoder in encoder-only and encoder-decoder model, and the decoder remains unchanged, so KV cache in the decoder remains unchanged and can be used simultaneously to complement our method in encoder.
>
> 3. Missing reference. It would be better to have a discussion on the difference with the previous [1] and concurrent [2] work that scales Transformer to 262k and one million, respectively.
>
> Thanks for the suggestion. In our paper, we discussed differences with some previous works, but yes, we are happy to include more discussion on these related works about Memorizing Transformers [1] and RMT [2].
>
> Both Memorizing Transformers and RMT follow a recurrent design and store the past context in an external memory module (memory with kNN lookup for Memorizing Transformers and compressed memory for RMT). They are good for autoregressive decoding settings, but lack the ability of bidirectional encoding. More importantly, similar to efficient Transformers with linear cost attention, both approaches do not try to reduce the linear cost (on sequence length) for FFN, so the computation might still be expensive for processing ultra long sequences. By sequentially processing one segment of input sequences at one time and truncating back-propagated gradients to the external memory, they avoid blowing up the GPU memory when processing ultra long sequences.
>
> On the other hand, our method seeks to reduce the overall cost (both self-attention and FFN) of processing ultra long sequences and processes the entire sequence simultaneously. We will include more discussions about the comparison to these works in our paper.
>
> [1] Memorizing Transformers. In ICLR 2022.
>
> [2] Scaling Transformer to 1M tokens and beyond with RMT. In arXiv.
>
> Thank you for your appreciation and feedback, and we will be glad to answer any additional questions.

---

> > ### Author Response · Authors · 2023-08-17
> >
> > Since there are only a few days left in the discussion period, it would be very helpful for us to know if our answers were satisfactory and if we can answer anything else. Any feedback is appreciated. Thank you!

---

> > > ### Comment · Reviewer_z4Ur · 2023-08-18
> > >
> > > Thanks for the rebuttal. The authors have addressed my concerns, and therefore I maintain the positive score.

---

> > > > ### Author Response · Authors · 2023-08-19
> > > >
> > > > Thank you, we appreciate your feedback.

---

### Official Review · Reviewer_JBnL · 2023-07-06

**Soundness:** 2 fair
**Presentation:** 3 good
**Contribution:** 3 good
**Rating:** 5
**Confidence:** 5

**Summary:**

This work introduces a method to scale up the supported number of tokens for transformer-based models. It prioritize the important/dominant tokens and compress those unimportant tokens during the processing. The conducted experiments demonstrate the effectiveness of the proposed method, and show that the proposed method can be directly incorporated into existing pretrained models with additional finetuning.

**Strengths:**

I think this work is trying to target an important problem, i.e., how to break through the limitation on the number of tokens to be processed. The proposed VIP-token centric compression is overall reasonable in terms of its high-level idea. The coducted experiments provide evidences for its efficiency benefits and performance superority when compared to the baselines with the same sequence length.

**Weaknesses:**

The targeted problem and the main story are quite attractive at the first glance. However, the proposed methods are not that solid and insightful as expected, making I feel some parts somewhat overclaimed. To name a few, for the high-level idea of this work, evaluating the importance of VIP-tokens should be one of major parts. However, the proposed method for evaluating the importance of difference tokens are based on some intuitive assumptions. These assumptions are task-specific or case-specific, affecting the generalization capacity of the proposed method in fact.

Besides the proposed method for evaluating the importance of tokens, other parts of the proposed method only introduces high-level idea deas and illustrates them with examples, without providing a rigorous modeling.

The experiments are conducted mainly on ultra long sequences. This matches with the major story, but in fact, prioritizing important tokens should be generally benefitial for a wide range of tasks using transformer-based models. The authors have not provides discussion or analysis on these.

**Questions:**

1. Are the proposed methods, which include but are not limited to the method for evaluating the importance of different tokens, generally applicable (e.g., for other tasks, or for other modalities)? If yes, pls explain more. If not, please analyze the limitations in detail and in-depthly.
2. The proposed compression and decompression are in fact implemented by matrix multiplication. How can you ensure the tokens of different importances (not referring to splitting as VIP or non-VIP) are compressed or decomposed as expected?
3. Why can placing VIP tokens in the front seats by re-ordering play the role of prioritizing them? How about repeating the VIP tokens (this is a more commonly used way for highlighting important tokens)? Pls compare these two, if possible.

**Limitations:**

The biggest problem is that this paper is quite attractive at the beginning, but the main parts, especially for the proposed methods, are somewhat disappointing since they are not that solid and insightful as expected. Pls see the weakness part and my questions for more detailed comments.

---

> ### Author Rebuttal · Authors · 2023-08-09
>
> 1. Evaluating the importance of different tokens is based on some intuitive assumptions. Assumptions are task-specific or case-specific.
>
> We are a little unclear about the concern “evaluating the importance of different tokens are based on some intuitive assumptions”. We will be grateful if the reviewer could clarify or suggest any additional experiments. We will answer the question below based on our current interpretation:
>
> Yes, this approach is based on intuition from experimental feedback.
>
> Only a small subset of tokens (VIP-tokens) are important for the final prediction. This is true for many tasks. For example, in classification tasks, only the representation of CLS token is used to make the prediction. The representations of other tokens are only used to aggregate their information into the CLS token in the intermediate layers. **The selection of VIP-tokens is task specific, but this selection can be made with some simple knowledge for most use cases.** (please also see reviewer 2SnC Q1 response)
>
> Only a small subset of tokens (important non-VIP tokens) in a long sequence will provide relevant information for the final prediction. If the assumption does not hold, we will see performance degradation. But we achieve good performance on a variety of downstream tasks which suggests that the assumption at least partially holds. **The selection of important non-VIP tokens is guided by VIP-tokens using our algorithm and does not require user input.** Based on approximation quality presented in the answer to reviewer xiLt’s Q2, our VIP centric compression does well in preserving important non-VIP tokens in the compressed sequence.
>
> 2. Only introduces high-level ideas and illustrates them with examples, without providing rigorous modeling.
>
> In the main text, we intentionally provide a high-level idea and toy examples (in section 3.3 and 3.4) for presentation purposes because introducing all low-level details will distract from the overall rationale of the approach and its effectiveness.
>
> Also, including all of these details seemed to make the paper unnecessarily dense and was distracting from the main message. To ensure that the ideas were accessible to a broad cross-section of the readership, we decided to move many of the technical details to the Appendix.
>
> A rigorous step-by-step derivation of our method is discussed in the Appendix section 2. The formal descriptions of the procedures introduced in section 3.3 and 3.4 are described in Algorithm 1 (page 6) and 2 (page 9) in the Appendix. In the main text, we give several pointers (at line 205, line 219, line 245, line 291) to the Appendix for the reader interested in such details. If there are specific sub-sections that will be valuable to include in the main text, we appreciate suggestions and will make every attempt to include in the main text. Thank you!
>
> 3. Authors have not provided discussion or analysis on other tasks/modalities that would be benefit from prioritizing important tokens.
>
> Yes, it is true that prioritizing important tokens is generally beneficial. As described in lines 379-387, our construction is most useful for ultra-long sequences. For other tasks with shorter sequences (< 4K), vanilla Transformer or efficient variants (Longformer, Big Bird, etc.) may be sufficient. Nonetheless, we can still test our method on other tasks. Please refer to our response to reviewer 2SnC Q3.
>
> 4. Why can placing VIP tokens in the front seats by reordering play the role of prioritizing them? How about repeating the VIP tokens?
>
> Thanks for this question. The goal of placing VIP tokens in the front seats is not to prioritize them. Instead, re-ordering makes the analysis, implementation and presentation of our method much clearer and simpler (line 167 in main text). In fact, placing VIP tokens at the end of the sequence can also serve the same purpose.
>
> As shown in the paper, by placing VIP tokens in the front, we can decompose the computation of a Transformer block into (a) computation for VIP tokens, (b) computation for non-VIP tokens, and (c) interaction between VIP tokens and non-VIP tokens for easier analysis. By reordering, our implementation also becomes easier.
>
> Why not repeat the VIP tokens? Suppose the sequence is [T0, T1, T2, …], and T0 is the VIP-token, then our interpretation is that after repeating (say once), the sequence will become [T0, T0, T1, T2, …]. Here, repeating the VIP-tokens means adjusting the attention weights such that the attention to T0 has higher weights. Our goal is to use VIP-tokens to guide compression of non-VIP tokens rather than let the non-VIP tokens pay higher attention to VIP-tokens.
>
> Please let us know if we misinterpreted the question.
>
> 5. Compression and decompression are implemented by matrix multiplication. How to ensure the tokens of different importances are compressed or decomposed as expected?
>
> We should clarify that we construct different compression matrices $S_c$ for different sequences. At a high-level, the compression matrix $S_c$ is constructed adaptively based on the different importance scores (VIP tokens’ attention scores to the compressed representation of segments) of non-VIP tokens. So, we ensure proper compression and decompression by properly constructing $S_c$ according to each input sequence.
>
> Note that in our algorithm, we do not need to perform matrix multiplication to obtain $S_cC$, rather we can obtain the rows of $S_cC$ while we are constructing the set J in Appendix Algorithm 1(see Appendix line 218-219). In visualization (Fig 3), the rows of $S_cC$ are the leaf nodes of the tree after growing the tree.
>
> For implementation, similarly, we use PyTorch operators including sorting (based on “different importances”), tensor sliding, and other operators to obtain the rows of $S_cC$.
>
> Thank you for your feedback, and we are happy to answer any additional questions.

---

> > ### Author Response · Authors · 2023-08-17
> >
> > Since there are only a few days left in the discussion period, it would be very helpful for us to know if our answers addressed your concerns and questions. Please let us know if we need to clarify anything else. Any feedback is appreciated. Thank you!

---

> > > ### Comment · Reviewer_JBnL · 2023-08-21
> > > **Thanks for your responses.**
> > >
> > > Thanks for your responses. Your interpretation for Q1 is correct, but I still think requiring task-specific knowledge for selecting VIP-tokens is kind of heuristic and may limit the wide application. Overall, your responses address most of my concerns. And I increase my score to borderline accept.

---

> > > > ### Author Response · Authors · 2023-08-21
> > > >
> > > > Thank you for the feedback. And we are happy that your concerns are mostly addressed. And thank you for raising the score!

---

### Official Review · Reviewer_2SnC · 2023-07-08

**Soundness:** 3 good
**Presentation:** 3 good
**Contribution:** 2 fair
**Rating:** 6
**Confidence:** 4

**Summary:**

VIP-token centric compression (VCC) aims to solve the quadratic attention complexity bottleneck in the self-attention component and the transformer encoder block. Roughly, it denotes some tokens as very important (VIP) and compresses the remaining sequence using a low-rank approximation. They benchmark on natural language data and do experiments to verify the time and space complexity from their theoretical analysis.

**Strengths:**

They achieve impressive accuracy and performance numbers. Strong results on question-answering tasks that require long context and small runtime. They scale to 128k tokens and are able to tackle NarrativeQA, although the results are not as strong.

The method is intuitive and the analysis and description of the method is clear.

They were able to get strong results with pretrained weights from existing transformer encoder models.

**Weaknesses:**

The idea of VIP tokens is similar to the global memory component of efficient transformers like ETC, BigBird, and Longformer. It suffers a similar problem that it's not clear which tokens are VIP.

Experiments are only done on encoders. They mention in passing that the method can be adapted to decoding, which seems to be more in the vogue these days.

Some aspects of the experiments like doing a different computation in the first 4 transformer layers make me think this method may be finicky and hard to apply in general.

**Questions:**

How does this method perform in other large context non-language tasks like images or audio or even synthetic tasks like Long Range Arena where the VIP token selection is less obvious?

It would be great to see decoder experiments?

**Limitations:**

Authors should be commended for addressing weaknesses of the method like the selection of VIP tokens and encoder-only aspect.

---

> ### Author Rebuttal · Authors · 2023-08-09
>
> Please refer to the global response for our response to the questions about decoder models.
>
> 1. The idea of VIP tokens is similar to the global memory component of efficient transformers like ETC, BigBird, and Longformer. It suffers a similar problem that it's not clear which tokens are VIP.
>
> It is true that selecting VIP-tokens requires some understanding of the tasks. However, we believe that a reasonable selection can be made with some generic knowledge for most tasks or use cases. For example, for question answering tasks, we just use questions as VIP-tokens. For classification tasks (for example, Long Range Arane benchmark shown below), we simply use CLS token as the VIP-token since only CLS token is used for final prediction. For masked language modeling, we use the masked tokens as the VIP-tokens since only these masked tokens are used for final prediction. For question answering, the question tokens (and candidate tokens for multi-choice QA) are used as the VIP-tokens since “they (1) are important to the specific task goals and (2) easily pre-identifiable by the user.” In the worst case, if there is no obvious token to be selected, we can prepend some learnable “latent” tokens or certain user commands (such as “summarize” for summarization tasks as we used in our experiments) and use them as VIP-tokens (in fact, CLS tokens can be thought of as these learnable tokens). We are happy to further emphasize this point in the paper.
>
> 2. Some aspects of the experiments like doing a different computation in the first 4 transformer layers make me think this method may be finicky and hard to apply in general.
>
> In our experiments, we use the first 4 transformer layers to process each 512-length non-overlapping segment of the input sequence into a reasonably good initial representation for our compression to operate on. Since there is no communication among segments, the downstream tasks cannot be solved by these first 4 transformer layers alone.
>
> We should clarify that our method also works in the setting where the compression is applied to all transformer layers. In our early evaluations, we also tested the setting with 12 x VCC transformer layers, but it leads to some performance degradation compared to the setting with 4 x short segment transformer layers + 8 x VCC transformer layers (for base models).
>
> Also, we note that unlike alternative methods that change the size of embedding matrices in the intermediate layers (such as gradually reducing sequence length as the input propagates to deeper layers), our method is easy when used as a drop-in replacement for vanilla transformer layers. This is because regardless of whether our compression is applied, similar to a Transformer block, the output of each layer is the same sized embedding matrix of the input sequence. Further, our implementation also makes it easier to use and experiment with since we only need to specify what type of computation (short segment transformer or VCC transformer) to use at each layer. Finally, our implementation works with arbitrary arrangements of computation as shown below:
>
> For example, the following is what we used for encoder-decoder model in experiment section:
>
> `encoder_layers = ["src.t5.models.layers.T5Block"] * 6 + ["src.t5.models.vcc_layers.VccT5Block"] * 18`
>
> We can also alternatively use these two types of layers in an alternating order:
>
> `encoder_layers = ["src.t5.models.layers.T5Block", "src.t5.models.vcc_layers.VccT5Block"] * 12`
>
> 3. How does this method perform in other large context non-language tasks like images or audio or even synthetic tasks like Long Range Arena where the VIP token selection is less obvious?
>
> While the focus of this work is on language tasks, the overall design does not limit its application to other tasks as long as the assumption (“a subset of tokens are disproportionately responsible for the model prediction” discussed in our paper and VIP-tokens can be reasonably selected) holds or partially holds for the tasks.
>
> After reading the reviewer’s suggestion, we also tested our method on the Long Range Arena (LRA) benchmark and obtained very promising performance among all baselines compared in MRA-attention [1] (We get slightly better performance than the top performing baselines presented in MRA-attention, but note that we are not trying to show that our method outperforms other baselines but to verify that our method can indeed be applied to other tasks.).
>
> Note that there are multiple implementations and hyperparameters used in the LRA benchmark, and comparisons across different implementations and hyperparameters is awkward. We use the same implementation and same hyperparameters as MRA-attention. Here, we present the LRA results of our method (average and 95% error bar calculated over 3 trials). Please refer to Table 5 in MRA-attention [1] for the LRA results of other baselines.
>
> ListOps: $37.3 \pm 0.5$
>
> Text: $65.3 \pm 0.2$
>
> Retrieval: $80.9 \pm 0.5$
>
> Image: $41.3 \pm 1.7$
>
> Pathfinder: $74.6 \pm 1.5$
>
> Avg: $59.9 \pm 0.9$
>
> The VIP token selection in LRA experiments is actually quite easy. We simply use the prepended CLS token as the only VIP token since it is responsible for the final model classification.
>
> Since LRA consists of a synthetic task (ListOps), language tasks (Text, Retrieval), and vision tasks (Image, Pathfinder), these results can serve as **preliminary evidence** indicating the potential applications of our method on other non-language tasks.
>
> [1] Multi Resolution Analysis (MRA) for Approximate Self-Attention
>
> Thank you for your supports, and we are happy to answer any additional questions.

---

> > ### Author Response · Authors · 2023-08-17
> >
> > Since there are only a few days left in the discussion period, it would be very helpful for us to know if our answers were satisfactory and if we can answer anything else. Any feedback is appreciated. Thank you!

---

### Official Review · Reviewer_FRn4 · 2023-07-10

**Soundness:** 3 good
**Presentation:** 3 good
**Contribution:** 3 good
**Rating:** 6
**Confidence:** 4

**Summary:**

The paper presents a new method to reduce the computational cost of transformer models on long input sequences. It proposes to use certain tokens (VIP tokens) that will be responsible for compressing the rest of the sequence into fewer tokens that can then be processed efficiently. In detail, a tree is constructed from the input features to the transformer block such that at each node we have the mean of the children of that node. To construct the feature sequence that will be processed, the averaged attention of the nodes of the tree with the VIP tokens is used to walk the tree until the leaves are reached. The reached nodes form the input sequence to the transformer block. Using the tree and the stored differences between the nodes and their children the new sequence is constructed after the transformer block. Thorough experiments show that the proposed attention achieves results on par with dense attention while being significantly faster to compute.

**Strengths:**

- Both using a set of tokens to generate the "clustering" of the input sequence as well as using the tree of differences to recreate the full sequence are very interesting and original methods.
- The experimental analysis is thorough and clearly shows that the proposed method works as well as dense attention in the examined benchmarks

**Weaknesses:**

- The main benefit of the method is the runtime efficiency, however the experiments do not describe adequately what this measured runtime is.
  1. It is mentioned that it is the time for a single sequence? How does the method scale wrt batch size?
  2. It is not clear whether it is the forward pass time or the forward/backward. How does this method scale for training?
- There is a lot of theoretical analysis that seems slightly unnecessary as it doesn't provide much clarity but rather confuses given that the main approximation that allows for the method is taken from [6] and is not analyzed further in this work.
- The proposed method cannot trivially be used in an autoregressive setting as it is really hard to efficiently compress N tokens while information flows only from the first to the N-th.

**Questions:**

My main questions have to do with the calculation of the runtime of the presented method.

Are the numbers for a single sequence? In that case how does the method scale with the batch size given that different samples will have different paths in the tree which will result in many out of order copies that may make the method slower.

In addition, is the calculated runtime for a forward pass or for a full training step?

**Limitations:**

The authors adequately discuss the limitations of their work.

---

> ### Author Rebuttal · Authors · 2023-08-09
>
> Please refer to the global response for our response to the questions about decoder models.
>
> 1. Clarification about how the runtimes are measured.
>
> Thanks for asking this question. The runtimes presented in the experiment section are measured runtimes of a complete training step (including both forward and backward). For each method, we use the largest batch size that can fit into a 80GB A100 and measure the average latency of 10 steps. Then, the average latency is divided by the batch size to get the estimated runtime for a single instance. Via this procedure, we seek to measure the peak efficiency of each method when the GPU is at the highest possible utilization. We will make this clear in the paper. Let us know if this clarification addresses the question.
>
> 2. How does the method scale with the batch size given that different samples will have different paths in the tree which will result in many out of order copies that may make the method slower.
>
> As noted in the answer above, the batch size is larger than 1. Each individual instance will have its own tree, but this does not make the method any slower (for single instance runtime).
>
> While in the main paper, the compression and decompression is represented in a matrix form $S_c$ and the pseudo inverse $S_c^{\dagger}$, in our algorithm (Figure 3 and Algorithm 1 and 2 in the Appendix), the mathematically equivalent $S_cC$ is obtained by gathering the leaf nodes of constructed tree (Figure 3 visualization) or obtained from the constructed set J (Algorithm 1 and 2 in the Appendix).
>
> Similarly, in our implementation, we use sorting, tensor sliding, average pooling and others, to obtain a mathematically equivalent $S_cC$ with higher efficiency. All of these operators are parallelized along the batch dimension natively in PyTorch, and will be available within the companion codebase for the paper.
>
> 3. There is a lot of theoretical analysis that seems slightly unnecessary as it doesn't provide much clarity but rather confuses given that the main approximation that allows for the method is taken from [6] and is not analyzed further in this work.
>
> As briefly discussed in Appendix 2.6, the method is inspired from [6], but [6] itself does not directly provide the approximation that we need. Since the readership is broad, we felt that it would be useful for a reader to check that the algorithm is sound (see how our analysis in Appendix is connected to Equation 12 in the main text via Claim 2.2 in Appendix). The remaining analysis, such as the one in Appendix 2.5, discusses the implications of applying this approximation for compressing the sequence. We are happy to use this suggestion to further improve the clarity of the analysis or if suggested, deemphasize it.
>
> [6] Multi Resolution Analysis (MRA) for Approximate Self-Attention
>
> We are happy for your appreciation of our work and will be glad to answer any additional questions.

---

> > ### Author Response · Authors · 2023-08-17
> >
> > Since there are only a few days left in the discussion period, it would be very helpful for us to know if our answers were satisfactory and if we can answer anything else. Any feedback is appreciated. Thank you!

---

### Author Rebuttal · Authors · 2023-08-09

We are grateful for all reviewers' feedback and support. Please let us know if there are any remaining doubts or questions, and we are happy to answer any additional questions.

1. Can our method be applied to decoders and decoder-only models?

As we discussed in the paper, although it is possible to apply our method for some decoder-only settings, applying the proposed method in an autoregressive setting is not straightforward, and the implementation is non-trivial. Preventing information leak of future tokens is one of the challenges.

The goal of the paper is to reduce the cost of attention **as well as the feedforward network (FFN) block without changing the internals of the transformer block** when dealing with ultra long sequences (which differentiates our work from the few recent ideas that deal with ultra long sequences). Based on the best feasibility of this goal, the scope of application is focused on the encoder and encoder-decoder settings.

For standard autoregressive decoder settings, it might be difficult to compress the token that is being generated with neighboring tokens. It is in fact challenging to reduce **the cost of FFN** since every generated token will most likely need a full calculation of FFN when it is being generated.

Having said that, there are still clear opportunities for decoder-only models, which are left as our future work.

- As discussed in Appendix 5, for this setting, we can apply this compression for reducing the cost of causal attention, but the efficiency gain will not be as significant as the encoder and the implementation will be more complicated when we incorporate it with the existing accelerations such as KV cache.

- In many cases, the current LLMs are not used directly to generate an ultra long text. Rather, the requirements for processing ultra longer context manifests due to the need to incorporate user input and previous LLM responses (like ChatGPT) or to incorporate search results (such as New Bing). In these cases, there is a prefix context, and LLMs will generate text based on the user prompt and prefix context. We note that our method can indeed be applied to compress the prefix context, and the user prompt and currently generated tokens will be the VIP tokens.

Based on the reviewers’ comments, we will include additional discussion regarding the limitations and these solutions for decoder-only models in the paper.

---

### Decision · Program_Chairs · 2023-09-21

**Decision:**

Accept (poster)

**Comment:**

This paper presents a new method to reduce the computational cost of transformer models on long input sequences. All reviewers recommended acceptance, based on merits such as a novel method (VIP-token centric compression) to improve efficiency of Transformers for long sequences, achieving good performance on several long-sequence tasks while being faster than compared baselines, and thorough experiments validate effectiveness. There are also some limitation pointed out such as the method currently only applies to encoder, not decoder models. Selection of VIP tokens requires some task-specific knowledge. FLOPs numbers missing. Runtime hardware-dependent. Some concerns on approximation error of compression technique, which warrants more analysis. Compatibility with relative positional encoding unclear. Differences with very related previous work not discussed. The authors are encouraged to take the feedback into account to improve the final revision.